# PDE and agent based simulation approaches to Ischemic Dermal Wound Closure

Teddy Lazebnik[1,2,☉*], Avner Friedman[3☉]

**1** Department of Information Systems, University of Haifa, Haifa, Israel, **2** Department of Computing, Jonkoping University, Jonkoping, Sweden, **3** Department of Mathematics, The Ohio State University, Columbus, Ohio, United States of America

☉ These authors contributed equally to this work.
\* lazebnik.teddy@gmail.com

## Abstract

Ischemic dermal wounds present a significant medical challenge, especially in the case where the wound does not close in an expected time, typically 30 days. We developed two very different mathematical models of symmetric flat wounds, one by Partial Differential Equations (PDE) and another by Agent-based Simulation (ABS) with some parameters taken from the PDE model. The models include the important role of keratinocytes who make 90% of the cells in the epidermis. We used both models to assess the effectiveness of oxygen therapy in wound closure for different levels $\alpha$ of ischemia; ischemia increases as $\alpha$ increases from 0 to 1. We found that (i) the decreasing profiles of the radius $R(t)$ of the open wound derived by the two models are in a high degree of agreement, and (ii) standard hyperbaric and topical oxygen therapies effectively achieve complete closure of the wound in expected time in cases where the ischemic level is not too high, i.e., $\alpha \leq 0.3$ under standard hyperbaric therapy and $\alpha \leq 0.5$ under continuous topical oxygen therapy. These findings provide a quantitative framework for evaluating ischemic wound healing and therapeutic interventions.

## 1. Introduction

Spatio-temporal mathematical models of biological processes, which take place in a domain with a known boundary, are commonly represented by a system of partial differential equations (PDEs). But when the boundary of the domain varies in time, some assumptions must be made on the dynamics of the boundary that will enable us to solve the PDE system simultaneously with the unknown boundary.

In some cases where these assumptions are not necessarily correct, an entirely different approach, known as agent-based simulation (ABS) may be more, or equally, useful. ABS is a stochastic model where, in a biological process, cells move in a grid-geometry, and proteins determine the dynamics of the environment. In ABS, no assumptions are imposed on the unknown boundary, but in order to derive "reliable"

**Data availability statement:** All relevant data are within the paper.

**Funding:** The author(s) received no specific funding for this work.

**Competing interests:** The authors have declared that no competing interests exist.

results, one must perform many repetitions of the simulation and then take their average.

In this paper, we use both methods (PDE and ABS) to address a biomedical problem and compare their respective conclusions. The problem is to determine the in-time closure of an ischemic dermal wound with or without therapy. This is an important medical problem, since wounds that remain open for a long time increase the risk of infection in the whole body.

The skin has three main layers: the dermis is the middle layer, the epidermis layer is above, and the hypodermis is below. The epidermis is the thinnest layer; it helps hydrate the body and protect it from damage. Most of the cells in the epidermis are keratinocytes, a highly specialized type of epithelial cells. The dermis is the thickest layer of the skin. It supports the epidermis by providing strength and flexibility, and its blood arteries transport (by diffusion) nutrients to the cells in the epidermis. The dermis also contains sweat glands, hair follicles, collagen and elastin, and nerve cells. The hypodermis (subcutaneous tissue) connects the skin to the muscles and bones of the body.

The healing process of dermal wounds is divided into four overlapping phases. In the first phase, clotting factors are delivered by platelets immediately after injury. In the second phase, called the inflammatory phase, platelets release growth factors (PDGF), which attract pro-inflammatory M1 macrophages to clear the inflammation in the open wound. In the next phase, called the proliferative phase, M1 macrophages polarize into anti-inflammatory M2 macrophages who, together with fibroblasts ($F$), begin the process of closing the wound. The expected time for normal wound closure is a few weeks, after which the phase of scar formation begins, and its completion may take many months. For definiteness, we assume that the expected closure of the wound is 30 days.

The closure of the wound in an expected time depends on a normal supply of oxygen by the peripheral artery. In ischemic wounds, where the peripheral artery in the dermis is impaired, resulting in oxygen deficiency, wound closure, in expected time, may not be completed without intervention by oxygen infusion. This situation was mathematically modeled in [1], in the special case of radially symmetric "flat" wounds, with radius $R(t)$, where the depth of the dermal wound is ignored. The wound healing model in [1] was represented by a PDE system of equations in the "partially healed tissue" (PHT), $R(t) < r < B$ where $B > R(0)$. The model included, in addition to macrophages and fibroblasts, also vascular endothelial growth factor (VEGF) which promotes angiogenesis, and density, $\rho$, of the extracellular matrix (ECM). However, in order to derive an equation for the unknown boundary, $r = R(t)$ of the open wound, several assumptions had to be made.

The first assumption in [1] was that PHT has the structure of upper convected Maxwell fluid with constant parameters independent of space, and the healing dynamic is quasi-static. This led to an equation for the ECM velocity $v = v(r, t)$ of the form:

$$\frac{1}{r}\frac{\partial}{\partial r}\left(r\frac{\partial v}{\partial r} - \frac{v}{r^2}\right) = \frac{\partial P}{\partial r},$$

(1)

where $P$ is the internal isotropic pressure associated with the ECM density ($\rho$) in PHT. The second assumption was that $P$ depends on $\rho$ as follows:

$$P = \begin{cases} \beta(\frac{\rho}{\rho_1} - 1) \text{ if } \rho \geq \rho_1 \\ 0 \text{ if } \rho < \rho_1 \end{cases} \quad , \tag{2}$$

for some parameters $\beta, \rho_1$. The third assumption was that ECM and all cells and cytokines move with velocity $v$, and, in particular, $dR/dt = v(R(t), t)$. Nevertheless, it was shown in [1] that the model simulations are in agreement with experimental results for ischemic dermal wounds.

It was subsequently shown, for this model, in [2], by mathematical analysis and simulations, that the time of wound closure increases if the oxygen supply from the boundary $r = B$ decreases.

The above model was later extended to include the depth of dermal wounds and wounds with non-spherical shapes. In [3], axially symmetric wounds were considered, and in [4], general 3d wounds were studied, by analysis, and with simulation in axially symmetric wounds. The model in [1] was extended in [5] to simulate wound healing and wound closure of chronic wounds in diabetes and obesity. We note, however, that all the above models do not include the role played by keratinocytes, who are the predominant cells in the epidermis [6,7].

Agent-based model (ABM) is a computational model for simulating the action and interactions of autonomous agents [8], while agent-based simulation (ABS) refers to computer implementation.

There are several models of wound healing based on ABM approach [9–12]; they represent activities between discrete cells, proteins, and other molecules, outside the wound, with cell migration of epithelial cells and other cells (e.g., immune cells, fibroblasts) into the wound.

ABS models have been used in cancer biomedicine; see comprehensive review in [13]. ABS models that use differential equations to qualify selected pathways, or communication between cells appeared in [14–17].

In this paper, we model the healing process only inside the wound, a region we call the partial healing wound (PHW). In the radially symmetric flat wound, $PHW(t) = \{R(t) < r < R(0)\}$. We are interested in the progress to wound closure of the epidermis layer, and, accordingly, we shall focus only on the proliferative phase of wound healing. Although dermal wounds may extend to the full thickness of the dermis, we shall consider here only the wound closure achieved by the keratinocytes, at the epidermal layer.

We first develop a PDE model of a radially symmetric flat wound with Eqs. (1−2) but include in the "flat wound" the epidermal layer whose thickness is very small, 0.07–0.15 mm [18]. The model variables include keratinocyte cells ($E$), which make up 90% of the epidermal cells [19], epidermal macrophages ($M$) [20], skin that fibroblasts ($F$) [21] which produce the ECM of the epidermis [22], density of ECM ($\rho$), VEGF ($V$), oxygen ($W$), TGF-$\beta$ ($T_\beta$), and wound area ($A(t) = \pi R^2(t)$).

Fig 1 is a network of interactions among the model variables; it will guide the development of the PDE model, and partially also of the ABS model.

## 2. Mathematical model

### 2.1. PDE model

The model variables are listed in Table 1 in densities with units of $g/cm^3$.

The model is represented by a system of partial differential equations in the region $PHW(t)$, based on the network in Fig 1.

We focus on the proliferation phase, where most $M1$ macrophages have already polarized into $M2$ macrophages at the initial time, $t = 0$; for simplicity, we do not include $M1$ explicitly in the model.

Fibroblasts and $M2$ macrophages are sensitive to hypoxia, and their proliferation is affected by the level of oxygen [23,24], which we take to be

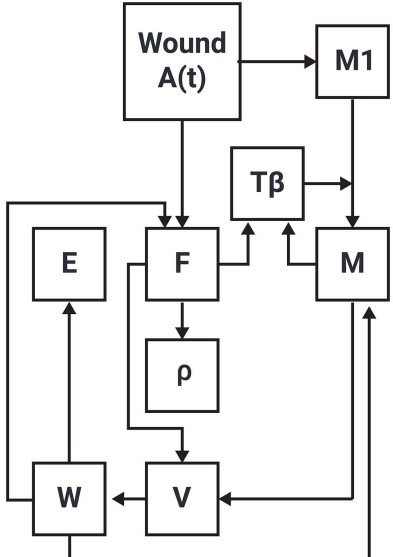

**Fig 1. Network of model's variables.** The arrows indicate activation, production, increasing, and enhancing.

**Table 1. A list of the model variables.**

| Variable | Definition |
| --- | --- |
| F | Fibroblasts |
| M | M2 macrophages |
| E | Keratinocyte cells |
| V | Vascular endothelial growth factor (VEGF) |
| W | Oxygen |
| $T_\beta$ | Transforming growth factor-beta (TGF-$\beta$) |
| $\rho$ | Extracellular matrix (ECM) density |
| $A(t)$ | Open wound area |

$$Q(W) = \frac{W}{W^0 + W},$$

(3)

where $W^0$ is the average oxygen concentration in human tissue.

**2.1.1. Equation for ECM density ($\rho$).** ECM is produced by fibroblasts, and this process is enhanced by TGF-$\beta$ [25,26]. We write the equation for $\rho$ as follows:

$$\frac{\partial \rho}{\partial t} + \nabla \cdot (v\rho) = \lambda_\rho Q(W)F(1 + \lambda_{\rho T_\beta} \frac{T_\beta}{K_{T_\beta} + T_\beta})(1 - \frac{\rho}{\rho_m}) - d_\rho \rho,$$

(4)

where $\lambda_\rho$, $\lambda_{\rho T_\beta}$ and $\rho_m$ are constants, and $d_\rho$ is the degradation rate of $\rho$.

The equation for each of the remaining species $X$ has the form

$$\frac{\partial X}{\partial t} + \nabla \cdot (vX) - \delta_X \nabla^2 X = F_X,$$

where $\delta_X$ is a diffusion coefficient, $v$ is the velocity, which in the case of radially symmetric flat wound satisfies Eqs. (1–2), and $F_X$ is determined by the network in Fig 1 (with $M$1 omitted).

**2.1.2. Equation for fibroblasts ($F$).** PDGF is released by damaged blood platelets in the wound, whose total mass is proportional to the wound area $A(t) = \pi R^2(t)$. PDGF stimulates logistic growth of fibroblasts at oxygen dependent rate $\lambda_F Q(W)$ [27]. Accordingly, we write the equation for $F$ as follows:

$$\frac{\partial F}{\partial t} - \nabla \cdot (vF) - \delta_F \nabla^2 F = A_F + \lambda_F A(t) Q(W) F (1 - \frac{F}{F_0}) - d_F F,$$

(5)

where $A_F$ is the source of fibroblasts, $d_F$ is the death rate of fibroblasts, and $F_0$ is the carrying capacity of $F$.

**2.1.3. Equation for M2 macrophages ($M$).** Blood monocytes are attracted to the wound and differentiate into pro-inflammatory M1 macrophages [28]. PDGF released from the wound stimulate growth of M1 macrophages [29]. During the proliferation phase, most M1 macrophages had already polarized to M2 macrophages, a process enhanced by TGF-$\beta$ [28]. For simplicity, we do not include M1 explicitly in the proliferation phase, and take the growth dynamics of M2 to mimic the growth dynamics of M1, so that

$$\frac{\partial M}{\partial t} - \nabla \cdot (vM) - \delta_M \nabla^2 M = A_M + \lambda_M A(t) Q(W) M (1 + \lambda_{MT_\beta} \frac{T_\beta}{K_{T_\beta} + T_\beta}) - d_M M,$$

(6)

where $d_M$ is the death rate of M; note that the growth of $M$ is oxygen dependent [30].

**2.1.4. Equation for Keratinocyte cells ($E$).** Keratinocytes make up 90% of the cells in the epidermis [19], and they play a role as structural cells that also exert important immune function [31]. Growth of keratinocytes cells depends on oxygen, hence

$$\frac{\partial E}{\partial t} - \nabla \cdot (vE) - \delta_E \nabla^2 E = \lambda_E Q(W) E (1 - \frac{E}{E_0}) - d_E E,$$

(7)

where $d_E$ is the death rate of $E$, and $E_0$ is the carrying capacity of $E$.

**2.1.5. Equation for VEGF ($V$).** VEGF is secreted by M2 macrophages and fibroblasts [29,32]. VEGF is lost in the process of angiogenesis. In this process, VEGF ligands to receptors on endothelial cells, and new blood capillaries are then formed near the wound, resulting in blood oxygen seepage into the wound. We view the proliferation of endothelial cells by VEGF as an "eating" process of VEGF by the endothelial cells, and use the Michaelis–Menten expression const. $V/(K_V + V)$ to represent the rate of loss of $V$. We write the equation for VEGF as follows:

$$\frac{\partial V}{\partial t} - \nabla \cdot (vV) - \delta_V \nabla^2 V = \lambda_{VF} F + \lambda_{VM} M - \hat{d}_V \frac{V}{K_V + V} - d_V V,$$

(8)

where the third term on the right-hand side represents a loss of $V$ in the process of angiogenesis, and $d_V$ is the degradation rate of $V$.

**2.1.6. Equation for oxygen ($W$).** Oxygen is increased by angiogenesis, when VEGF ligands to receptors on endothelial cells. Due to limited receptor recycling time, we model this increase in oxygen by the Michaelis–Menten expression $\lambda_{WV} V/(K_V + V)$, for some parameter $\lambda_{WV}$. We write the equation for oxygen as follows:

$$\frac{\partial W}{\partial t} - \nabla \cdot (vW) - \delta_W \nabla^2 W = (A_W + \lambda_{WV} \frac{V}{K_V + V})(1 - \alpha) - d_W (F + M + E) W,$$

(9)

where oxygen is supplied by blood cells at rate $A_w$, it is enhanced by VEGF at rate $\lambda_{WV}$, and is consumed by cells $F$, $M$, and $E$. The parameter $\alpha$ quantifies the level of ischemia, $0 \leq \alpha \leq 1$; when $\alpha$ increases from 0 to 1, the ischemic level increases from non-ischemia to total ischemia.

**2.1.7. Equation for TGF-$\beta$ ($T_\beta$).** TGF-$\beta$ is produced by fibroblasts [33] and M macrophages [34]. Hence,

$$\frac{\partial T_\beta}{\partial t} - \nabla \cdot (v T_\beta) - \delta_{T_\beta} \nabla^2 T_\beta = \lambda_{T_\beta F} F + \lambda_{T_\beta M} M - d_{T_\beta} T_\beta, \tag{10}$$

where $d_{T_\beta}$ is the degradation rate of $T_\beta$.

**2.1.8. Equation for $R(t)$.** We assume that the wound boundary decreases with the velocity $v$ of the ECM:

$$\frac{dR(t)}{dt} = v(R(t), t). \tag{11}$$

In the PDE model of a radially symmetric flat wound, Eqs. (1–11) hold in the partially healed wound (PHW) $R(t) \leq r \leq R(0)$, $t > 0$. In order to simulate the PDE system, we need to assume boundary conditions on the moving boundary $r = R(t)$ and the external boundary, $r = R(0)$.

## 2.2. Boundary conditions

For Eq. (1) we take

$$v = 0 \text{ on } r = R(0), \quad \frac{\partial v}{\partial r} = P \text{ on } r = R(t). \tag{12}$$

For oxygen we take

$$(1 - \alpha)(W - W^0) + \alpha \frac{\partial W}{\partial r} = 0 \text{ on } r = R(0), \quad \frac{\partial W}{\partial r} = 0 \text{ on } r = R(t) \tag{13}$$

where $\alpha$ is the parameter that quantifies the level of ischemia.

We denote by $X^0$ the average density, in health, of any species $X$ of cells or proteins, and take

$$(1 - \alpha)(X - X^0) + \alpha \frac{\partial X}{\partial r} = 0 \text{ on } r = R(0), \quad \frac{\partial X}{\partial r} = 0 \text{ on } r = R(t) \quad \text{for } X = F, M, E, \tag{14}$$

$$X = X^0 \text{ on } r = R(0), \quad \frac{\partial X}{\partial r} = 0 \text{ on } r = R(t) \text{ for } X = V, T_\beta, \tag{15}$$

and

$$\rho(R(0), t) = \rho_0 \text{ for all } t > 0. \tag{16}$$

The PDE system takes place in the region $\{R(t) \leq r \leq R(0)\}$, $t > 0$, and we take $R(0) = 1$ cm.

From Eqs. (9) and (11), we see that in the case of $\alpha = 1$ (total ischemia), $W = 0$, hence $Q(W) = 0$, $\rho = 0$, $E = 0$, $V = 0$, and $R(t) = R(0)$ for all $t \geq 0$; the wound will not begin to heal without treatment.

## 2.3. Parameters estimation

**2.3.1. Steady state in health.** We denote the healthy steady (or average) state of species $X$ by $X^0$, and assume that in steady state $\frac{X}{K_X+X} = 1/2$ where $K_X$ is the half-saturation of $X$; hence $K_X = X^0$.

The thickness of the epidermis of the human body is $0.07 - 0.15$ mm [18]; we take an average of 0.01 cm. The epidermis has 4 layers of stratum basale [35], and each layer contains 26–45 layers of keratinocyte cells [7]; we take an average of 30 layers. Each layer has 2500–5000 cells in $cm^2$ [19]; we take an average number of 4000 cells. Hence, the number density of keratinocytes is

$$\frac{1}{0.01}4 \cdot 30 \cdot 4000 = 4.8 \cdot 10^7 \text{ in } cm^3.$$

The size of a keratinocyte cell is 10–15 $\mu m$; hence its flat area is less than $10^2 - 15^2$ $\mu m^2$. But the vertical dimension is smaller than 0.01 divided by the number of the keratinocyte layers, $(0.01/120)cm = 1/1.2\mu m$. We accordingly take the volume of a keratinocyte cell to be $(10\mu m)^3 = 10^{-9}cm^3$. Assuming that 1 $cm^3$ full of cells has a mass of 1g, we get the density of keratinocytes in the epidermis to be

$$E^0 = 4.8 \cdot 10^7 \cdot 10^{-9} = 4.8 \cdot 10^{-2} \text{ } g/cm^3.$$

There are 2100–4100 fibroblasts in $mm^3$ of the mid-dermis [36], and 2000–4000 macrophages in $mm^3$ of the mid-dermis [37]. Since 90% of the cells in the epidermis are keratinocytes [19], we assume that the remaining 10% are macrophages and fibroblasts, in equal numbers. Assuming that the volume of each of these cells is $(10\mu m)^3$, we get

$$F^0 = M^0 = 5/100E^0 = 2.4 \cdot 10^{-3} \text{ } g/cm^3.$$

In skin of healthy mice, the density of VEGF is 150 pg/mg [38] (Fig 7). Assuming that the mass of 1 $cm^3$ of skin tissue is 1g, we get

$$V^0 = 1.5 \cdot 10^{-7} \text{ } g/cm^3.$$

In healthy skin the density of $T_\beta$ is 30–39 $pg/mm^3$ [39]; taking the average, we get

$$T_\beta^0 = 3.5 \cdot 10^{-8} \text{ } g/cm^3.$$

The concentration of oxygen is given by the formula (in text, section "Materials and Methods" of [40]): $W^0 = P_{O_2} \cdot \alpha_{tissue}$, where $P_{O_2} = 100M$/mmHg is the oxygen pressure in arterial blood and (from Table 3 in [40]) $\alpha_{tissue} = 1.25 \cdot 10^{-6}$ mmHg is the oxygen solubility in the tissue; here $M = 10^{-3}mol/cm^3 = 32 \cdot 10^{-3} g/cm^3$. Hence, $W^0 = 1.25 \cdot 32 \cdot 10^{-6} = 4 \cdot 10^{-6}g/cm^3$.

The ECM density is 3–4% of the dry weight of tissue [41]. Since the epidermis contains 70% water [42], we take $\rho_0 = 0.06g/cm^3$. We also take $\rho_m = 1.1\rho_0 = 0.066g/cm^3$, $\rho_1 = 0.2\rho_0 = 0.012g/cm^3$, and $\beta = 2.72 \cdot 10^{-3}/d$.

**2.3.2. Death and degradation rates.** The death/degradation rate $d_X$ of species $X$ is determined by the half-life $t_{1/2}(X)$ of $X$: $d_X = log(2)/t_{1/2}(X)$. From the half-life (or average of half-lives) in previous papers, get: $d_F = 0.02/d$ [43], $d_M = d_{M_2} = 0.099/d$ [44], $d_E = 0.0577/d$ [45], $d_V = 16.5/d$ [46], $d_{T_\beta} = 495/d$ [47], and $d_\rho = 0.37/d$ [1].

**2.3.3. Diffusion coefficients.** From previous work, we take the estimate $\delta_F = \delta_M = \delta_E = 8.64 \cdot 10^{-7}cm^2/d$ [48]; although this estimate is very rough, it seems to have the correct order of magnitude. The remaining diffusion coefficients were estimated more precisely as follows: $\delta_V = 8.64 \cdot 10^{-2}cm^2/d$ [49], $\delta_W = 2cm^2/d$ [50], $\delta_{T_\beta} = 7.1 \cdot 10^{-2}cm^2/d$ [51].

## 2.4. Steady state in health

We take $K_V = V^0 = 1.5 \cdot 10^{-7} g/cm^3$, $K_{T_\beta} = T_\beta^0 = 3.5 \cdot 10^{-8} g/cm^3$, and the carrying capacity of $F$ and $E$ to be $F_0 = 2F^0 = 4.8 \cdot 10^{-3} g/cm^3$ and $E_0 = 2E^0 = 9.6 \cdot 10^{-2} g/cm^3$.

In steady state of health, $A(t) = 0$ and $Q(W) = 1/2$, and Eqs. (5–10) take the following form:

$$A_F = d_F F^0 = 4.8 \cdot 10^{-5} g/cm^3 \cdot d, \; A_M = d_M M^0 = 23.76 \cdot 10^{-2} g/cm^3 \cdot d, \; \lambda_E = 4d_E = 23.08 \cdot 10^{-2}/d,$$

$$\lambda_{VF} F^0 + \lambda_{VM} M^0 = 0.5\hat{d}_V + d_V V^0, \; A_W + 0.5\lambda_{WV} = d_W(F^0 + M^0 + E^0)W^0, \; \lambda_{T_\beta F} F^0 + \lambda_{T_\beta M} M^0 = d_\beta T_\beta^0.$$

We assume that $\lambda_{VF} F^0 = \lambda_{VM} M^0$, $0.5\hat{d}_V = d_V V^0$, and conclude that

$$\lambda_{VF} = d_V V^0/F^0 = 16.5 \cdot 1.5 \cdot 10^{-7}/(2.4 \cdot 10^{-3}) = 1.031 \cdot 10^{-3}/d, \; \lambda_{VM} = d_V V^0/M^0 = 1.031 \cdot 10^{-3}/d,$$

$$\hat{d}_V = 2d_V V^0 = 33 \cdot 1.5 \cdot 10^{-7} = 4.95 \cdot 10^{-6} g/cm^3 \cdot d.$$

We assume that $d_W = 0.2/d$ and $A_W = 0.5\lambda_{WG}$, and find that $A_W = 0.5 \cdot 0.2 \cdot 5.28 \cdot 10^{-2} = 5.28 \cdot 10^{-3} g/cm^3 \cdot d$ and $\lambda_{WV} = 11.616 \cdot 10^{-3} g/cm^3 \cdot d$ (somewhat larger than $2A_w$). Finally, we assume that $\lambda_{T_\beta F} F^0 = \lambda_{T_\beta M} M^0$ and conclude that $\lambda_{T_\beta F} = 0.5d_{T_\beta} T_\beta^0/F^0 = 0.5 \cdot 495 \cdot 3.5 \cdot 10^{-8}/(2.4 \cdot 10^{-3}) = 3.608/d$, $\lambda_{T_\beta M} = 0.5d_{T_\beta} T_\beta^0/M^0 = 3.608/d$.

We take $\lambda_{\rho T_\beta} = 1$, and from the steady state of Eq. (4) we get:

$$\lambda_\rho \cdot 0.5 \cdot F^0 \cdot 1.5 \cdot 0.1/1.1 \cdot 0.5 = d_\rho \rho_0 = 0.37 \cdot 0.06,$$

so that $\lambda_\rho = 1.356 \cdot 10^2/d$.

**2.4.1. Parameters associated with $A(t)$.** We assume that in Eq. (5), $\lambda_F A(t)Q(W)F(1 - \frac{F}{F_0}) = \gamma_1 d_F F$ at $t = 0$ for some parameter $\gamma_1 = 3.2$. Since $R(0) = 1$ cm, $A(0) = \pi$, so that $\lambda_F \pi/4 = 3.2d_F = 0.064$; hence $\lambda_F = 8.154 \cdot 10^{-2}/cm^2 \cdot d$.

We take $\lambda_{MT_\beta} = 1$ in Eq. (6), and assume that $\lambda_M A(t)Q(W)M(1 + \frac{T_\beta}{K_{T_\beta} + T_\beta}) = \gamma_2 d_M M$ at $t = 0$; for some parameter $\gamma_2 = 1.7$. Assuming also that $T_\beta = T_\beta^0$ at $t = 0$, we get $\lambda_M \cdot \pi/2 \cdot 3/2 = 1.7d_M = 1.683 \cdot 10^{-1}$; hence $\lambda_M = 7.14 \cdot 10^{-2}/cm^2 \cdot d$.

## 3. ABS model

We define a uniform grid in two-dimensional space with $x$, $y$ axes, constructed using lines separated by a mesh size of $\Delta$. The set of centers of the resulting squares is denoted by $\mathbf{N}^2$. The Manhattan distance between two points $(x_1, y_1)$ and $(x_2, y_2)$ in $\mathbf{N}^2$ is given by the sum of the absolute differences of their coordinates $|x_1 - x_2| + |y_1 - y_2|$. The squares adjacent to a square centered at $(a, b)$ are the 8 squares with centers at $(a + i, b + j)$, where $i, j$ take values from $\{-1, 0, 1\}$. We model agents as cells, with each square can accommodate at most one cell, positioned at the square's center.

In agent-based simulation (ABS) based on the PDE model (Eqs. (1–16)), the agents are cells from $F$, $M$, and $E$, and the environment is associated with VEGF ($V$), oxygen ($W$), and TGF-$\beta$ ($T_\beta$). In setting up the ABS model, we assume that all the cells arrive from the boundary of the wound $r = R(0)$. We refer to the distribution of agents as the "geometry" of the model and assume any initial geometry.

Formally, an agent is defined by five parameters $(\tau, \bar{x}, \psi, \xi, p)$: $\tau$ is the cell type, with $\tau \in \{F, M, E\}$ in our specific model; $\bar{x}$ is the center of the square where the cell is located; $\psi$ is the lifespan of the cell; $\xi$ is the inner clock of the cell (in minutes); and $p$ is the non-zero pressure vector that represents the force applied to the agent by other agents to move within the geometry. In the simulations, we take the parameters for the ABS model from Table 1, but include velocity and diffusion in a different way than in the PDE model.

Following the ABS framework [52,53], we define three operators: spontaneous ($I_s$), agent-agent ($I_{aa}$), and agent-environment ($I_{ae}$). Given an initial geometry at time $t_0 = 0$, we run the operators $I_s$, $I_{aa}$, and $I_{ae}$ successively at times $t_1, t_2, \ldots, t_n, \ldots$ with equal time steps $t_n - t_{n-1} = \Delta t$ for all $n$, where $\Delta t = 1$ minute.

## 3.1. Operator $I_s$ (spontaneuos dynamics)

The life-span of $X \in \{F, M, E\}$ cells is derived from the equation $\frac{dX}{dt} = -d_X X$, or $X(t) = X(0)e^{-d_X t}$. Then, full life-span $\int_0^\infty X(t)dt = 1$ means that $\int_0^\infty d_X e^{-d_X t} dt = 1$, and the discrete probability $\psi$ is given by the exponential distribution: $\{\overline{d_X} e^{-d_X n}, n = 1, 2, \ldots\}$, $\overline{d_X} = 1/(e^{d_X} - 1)$ in units of days.

We set $t_n = t$ and $t_{n+1} = t + 1$. For any cell type $x \in X(t)$, if $\xi \geq \psi$ then we eliminate $x$, while if $\xi < \psi$ then we increase the cell inner clock time to $\xi + 1$; see Algorithm 1, lines 1–21.

For $X \in \{F, M, E\}$, we denote by $|X(t)|$ the number of cells in $X(t)$. For any positive real number $N$, we denote by $\lfloor N \rfloor$ the largest integer $\leq N$. We compute the number of cells to be added for all three cell types; see Algorithm 1, lines 22–24. All new cells are introduced at the boundary $r = R(0)$, endowed with random life-span from their exponential distribution, and pressure vector $p$ pointing inward the wound. If the location of a new cell on $r = R(0)$ was occupied by another cell, that cell is pushed over to adjacent location, determined by its pressure $p$. If the first push of a cell ends in a location already occupied by another cell, that cell is pushed by its pressure $p$ forward, and this pushing process continues until the last push ends at an unoccupied location. We performed the pushing process first with all $E$ cells, then with all $M$ cells, and finally with $F$ cells, as briefly indicated in Algorithm 1, lines 25–31.

In order to compute an approximation for the wound's radius at some time $t$, we define the size of the region unfulfilled by cells at time $t$ to be $A(t)$ and define the radius $R(t)$ by $\pi R(t)^2 = A(t)$ or $R(t) = \sqrt{A(t)/\pi}$.

## Algorithm 1 Spontaneous Dynamics ($I_s$) at time $t$

```
1: for each fibroblast cell in fibroblast cells (f ∈ F(t)) do
2:   if f.ξ ≥ f.ψ then
3:     Eliminate fibroblast cell (f)
4:   else
5:     f.ξ ← f.ξ + 1
6:   end if
7: end for
8: for each macrophage cell in macrophage cells (m ∈ M(t)) do
9:   if m.ξ ≥ m.ψ then
10:    Eliminate macrophage cell (m)
11:   else
12:     m.ξ ← m.ξ + 1
13:   end if
14: end for
15: for each keratinocyte cell in keratinocyte cells (e ∈ E(t)) do
16:   if e.ξ ≥ e.ψ then
17:     Eliminate keratinocyte cell (E)
18:   else
19:     e.ξ ← e.ξ + 1
20:   end if
21: end for
22: |F(t)^{new}| ← ⌊A_F + λ_F πR(t)^2 Q(Ŵ) · |F(t)| · (1 − |F(t)|/F_0)⌋
23: |M(t)^{new}| ← ⌊A_M + λ_M πR(t)^2 Q(Ŵ) · |M(t)| · (1 + λ_{MT_β} |T_β(t)|/(K_{T_β} + |T_β(t)|))⌋
24: |E(t)^{new}| ← ⌊λ_E Q(Ŵ) · |E(t)| · (1 − |E(t)|/E_0)⌋
25: Initialize new cells stack ← ∅
26: for n ∈ {E, M, F} in priority order do
27:   for i = 1 to |n(t)^{new}| do
28:     Add new n-cell to the boundary R(0)
```

```
29:     Apply inward pressure p, displacing lower-priority cells inward
30:   end for
31: end for
```

### 3.2. Operator $I_{aa}$ (agent-agent)

This operator is empty for our simulation as the three cell types are interacting with each other through the environment rather than directly.

### 3.3. Operator $I_{ae}$ (agent-environment)

At the beginning of the simulation ($t = t_0$), oxygen ($W$), VEGF ($F$), and TGF-$\beta$ ($T_\beta$) are divided in an equally distributed manner to all locations in the geometry, such that each square obtains the same number $|W(0)|$, $|V(0)|$, and $|T_\beta(0)|$, respectively. Next, following Algorithm 2, for each iteration, under the agent-environment operator ($I_{ae}$), oxygen is introduced uniformly to the geometry at rate $(A_W/S + 0.5\lambda_{WV})(1 - \alpha)$ where $S$ is the total number of locations in the geometry, and consumed by all the cells ($E$, $F$, $M$). In addition, fibroblasts ($F$) generate VEGF ($V$) and $T_\beta$ in the locations they are present at rates $\lambda_{VF}$ and $\lambda_{T_\beta F}$, respectively. In a similar manner, macrophages ($M$) generate VEGF ($V$) and $T_\beta$ in the locations they are present at rates $\lambda_{VM}$ and $\lambda_{T_\beta M}$, respectively. Then, for each location in the geometry, the new amount of a free $I \in \{W, T_\beta\}$ is obtained using the following formula of diffusion with a degradation coefficient $d_i$:

$$I_{i,j}(t + 1) = (1 - d_I)I_{i,j}(t) + \sum_{i' \in \{i-1, i+1\}} I_{i',j}(t) + \sum_{j' \in \{j-1, j+1\}} I_{i,j'}(t), \tag{17}$$

where $I_{i,j}$ stands for the amount of the free $I$ in location ($i,j$). The decay of $V$ is $-\hat{d}_V V/(K_V + V) - d_V V$, and for simplicity we take it to be $2d_V V$. Than, the diffusion and decay of $V$ is as follows:

$$V_{i,j}(t + 1) = (1 - 2d_V)V_{i,j}(t) + \sum_{i' \in \{i-1, i+1\}} V_{i',j}(t) + \sum_{j' \in \{j-1, j+1\}} V_{i,j'}(t). \tag{18}$$

**Algorithm 2 Agent-Environment Interactions ($I_{ae}$) at time $t$**
```
1:  for each location, l ∈ S, in the geometry do
2:      W_l ← W_l + A_W/S + 0.5λ_WV
3:  end for
4:  for each macrophage cell in macrophages cells (m ∈ M(t)) do
5:      V_{m.x̄} ← V_{m.x̄} + λ_VM
6:      T_{β m.x̄} ← T_{β m.x̄} + λ_{T_β M}
7:      W_{m.x̄} ← W_{m.x̄} - d_W
8:  end for
9:  for each fibroblast cell in fibroblast cells (f ∈ F(t)) do
10:     V_{f.x̄} ← V_{f.x̄} + λ_VF
11:     T_{β f.x̄} ← T_{β f.x̄} + λ_{T_β F}
12:     W_{f.x̄} ← W_{f.x̄} - d_W
13: end for
14: for each keratinocyte cell in keratinocyte cells (e ∈ E(t)) do
15:     W_{e.x̄} ← W_{e.x̄} - d_W
16: end for
17: Diffuse and decay VGEF (V) in the geometry using Eq. (17)
18: Diffuse and decay oxygen (W) in the geometry using Eq. (17)
19: Diffuse and decay TGF-β (T_β) in the geometry using Eq. (17)
```

## 4. Results

### 4.1. Computational method

The PDE model takes a second-order and nonlinear form with a free boundary spherical geometric configuration. We solve it numerically using the Runge-Kutta method [54]. Here, the boundary is updated at each step of the Runge-Kutta method. The free boundary is moved from one step to the next by updating the position ($x$) based on Eq. (11) (with the value of $v$ from the previous step) and the numerical time step $h$. This process involves evaluating $R(t)$ at the boundary point and then shifting the cell distribution in the numerical grid accordingly.

In the ABS model, for grid side $A(t)$, we define $R^2(t) = A(t)/\pi$ and we use this $R(t)$ to compare with the $R(t)$ of the PDE model.

All the numerical analysis in this study was performed using the Python programming language [55].

### 4.2. Wound closure without therapy

Fig 2 shows the radius of the wound ($R(t)$) for 30 days for $\alpha = 0, 0.1, 0.2, \ldots, 0.9$ for the PDE and ABS models. Due to the stochastic nature of the ABS model, the results for this model are shown as the mean ± standard deviation of $n = 100$ simulations. In the non-ischemic case ($\alpha = 0$), wound closure is complete already after 18 days. In the case of extreme ischemia ($\alpha = 0.9$), the wound does not close, and $R(30) \sim 0.8R(0)$. The coefficient of determination ($R^2$) that measures the goodness of fit between the PDE and ABS simulations averaged across the different values of $\alpha$ is $R^2 = 0.913$, which indicates that both models highly agree with each other in representing wound closure profiles.

Fig 3 shows the Keratinocytes cells's density in the wound ($E(t)$) for 30 days for the cases in $\alpha = 0, 0.1, 0.2, \ldots, 0.9$ for both the PDE and ABS models. Due to the stochastic nature of the ABS model, the results are shown as the mean ±

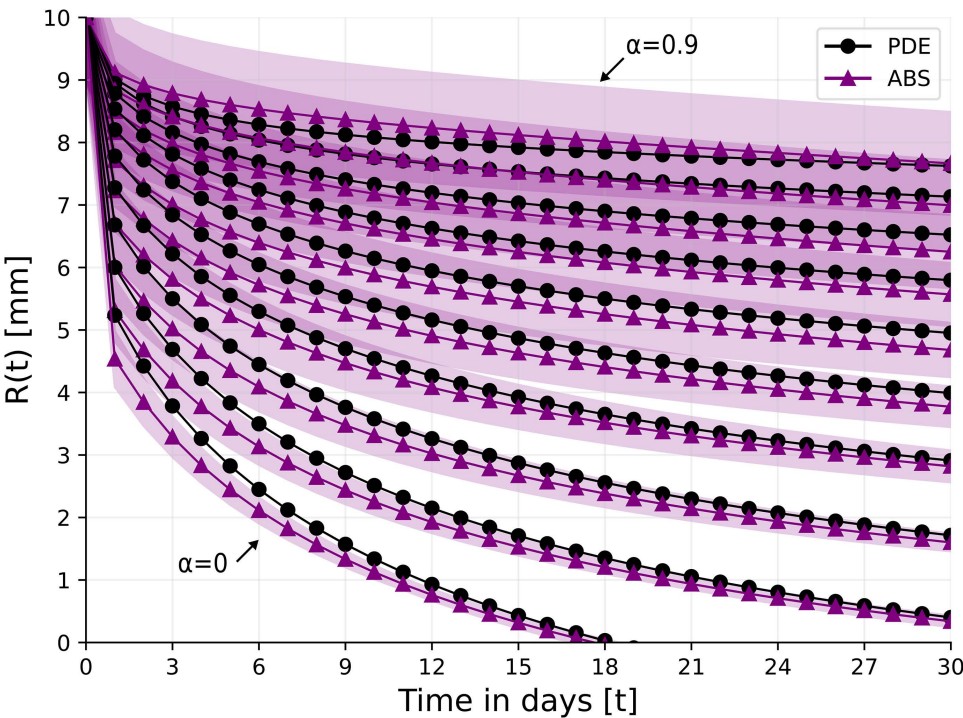

**Fig 2. Wound's radius over the course of 30 days for different levels of ischemia, $\alpha$ = 0, 0.1, 0.2,…, 0.9.** For the ABS model, the results are shown as the mean ± standard deviation of $n$ = 100 simulations.

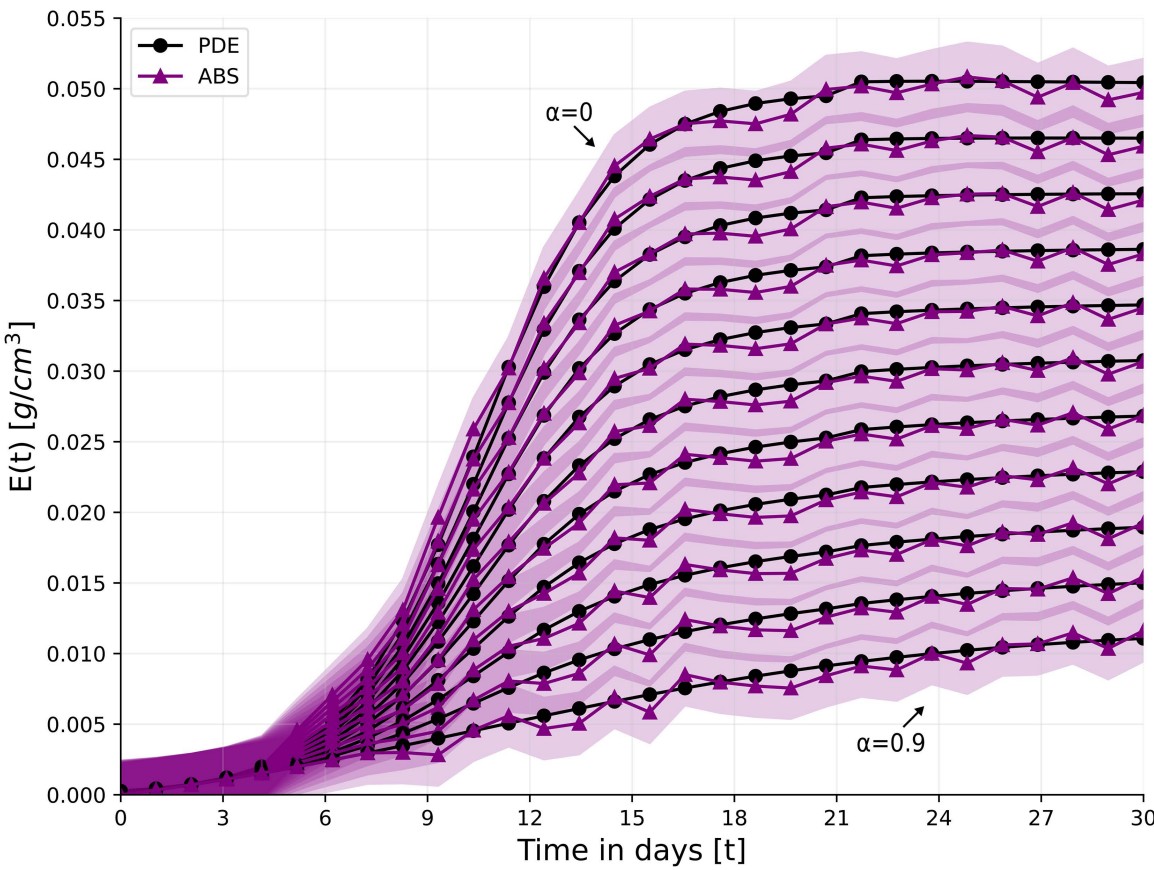

**Fig 3. Keratinocytes cells' average density in the wound over time for $\alpha$ = 0, 0.1, 0.2,···, 0.9.**

standard deviation of $n = 100$ simulations. In the non-ischemic case ($\alpha = 0$), the average density $E(t)$ is increasing until day 20, soon after wound closure is complete, and $E(20) \sim 4.96 \cdot 10^{-2} g/cm^3$, which is the keratinocyte cells density in the epidermis, in health. In the case of extreme ischemia ($\alpha = 0.9$), $E(t)$ is increasing in time but $E(30)$ is just slightly over $1.0 \cdot 10^{-2} g/cm^3$.

### 4.3. Comparison with experimental data

In vivo experiments with domestic white pig conducted in [56], identical wounds were developed in the healthy skin region and in the previously prepared ischemic skin region. Fig 3 in [56] shows the profile of the percentage of the initial wound radius for 20 days in both cases. Note that when a wound is developed, it dilates for the first few days before closure begins, as seen in [56] Fig 3. Fig 4A is taken from [56] Fig 3. Fig 4B shows the comparison between our simulations and Fig 4A; since the initial dilation is not included in our model, we start the comparison from day 3. We took $R(0) = 0.2$ cm as in [56] and computed, for each $0 \leq \alpha < 1$, the percentage of initial wound radius, for days 3, 4, ···, 19, 20. We then, connected these values linearly as done in Fig 4A. In the non-schematic case ($\alpha = 0$), we found that the measure of fitness ($R^2$) between the curve derived by the model and the curve in Fig 4A is $R^2 = 0.945$. In the ischemic case, we used the gradient descent method and least mean square to find $\alpha$ that yields the best fit to Fig 4A. We found that with $\alpha = 0.467$, the measure of fitness is $R^2 = 0.892$.

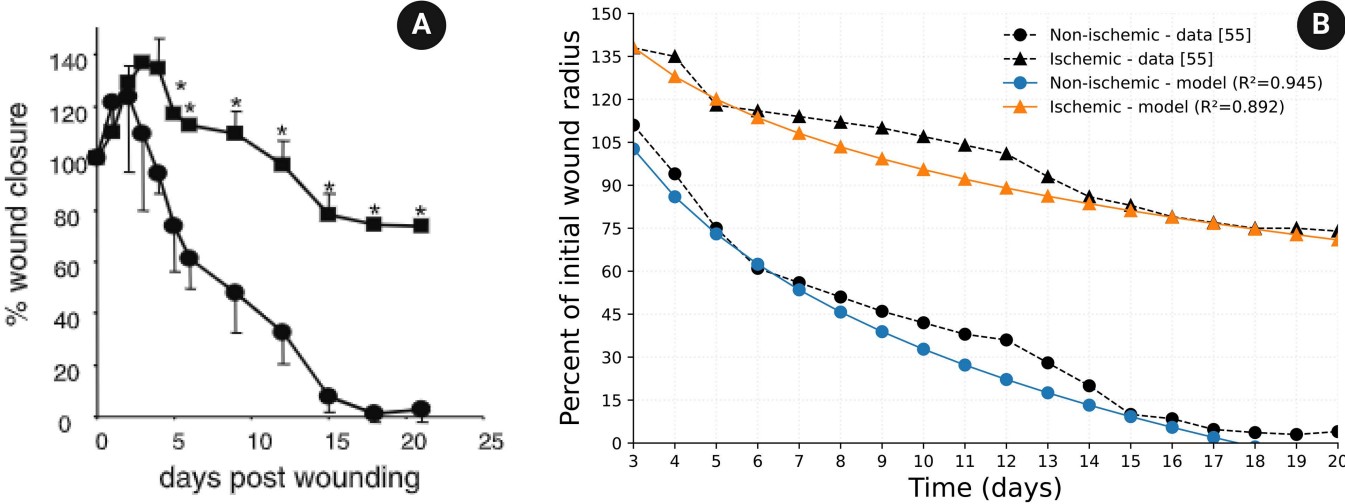

**Fig 4. Comparison between the model simulations and Fig 3 in [56].** Fig 4A is taken from Fig. 3 in [56]. Fig. 4B shows a comparison between the model's simulations and Fig. 4A in the non-ischemic case, and in the ischemic case with α = 0.467.

## 4.4. Oxygen therapy

There are two general approaches to oxygen therapy in ischemic wound healing: Hyperbaric Oxygen therapy (HBOT) and topical oxygen therapy (TOT).

In HBOT, a patient enters a special chamber, for 2 hours daily, to breathe pure oxygen in pressure levels of 1.5 to 3 times higher than oxygen pressure in air [57,58]. The high pressure of oxygen increases the systemic oxygen in the plasma, which then circulates to tissues and helps drive oxygen directly into the damaged tissue [59]. The increased oxygen pressure on the tissue surrounding the wound also begins to decrease the level of ischemia after 6–8 days, and, between 18–23 days, the number of blood vessels reaches 80% of normal tissue [60].

We model this decrease in ischemia by decreasing the initial parameter $\alpha = \alpha(0)$ to $\alpha(t)$, where $\alpha(t) = \alpha(0)$ if $t < 10$ days and $\alpha(t) = \alpha(0) - (\alpha(0) - 0.1) \cdot (t - 10)/20$ if $10 \leq t \leq 30$ days, and we modify Eq. (9) as follows:

$$\frac{\partial W}{\partial t} - \nabla \cdot (vW) - \delta_W \nabla^2 W = \left(A_W + \lambda_{WV} \frac{V}{K_V + V}\right)(1 - \alpha(t)) - d_W(F + M + E)W + \gamma_W h(t),$$ (19)

where $\gamma_W = \gamma_W^0 A_W(1 - \alpha(t))$, such that $\gamma_W^0 \in [1.5, 3]$ and

$$h(t) = \begin{cases} 1, & \text{if } 1 < t < 3 \text{ hours} \\ 0, & \text{elsewhere} \end{cases}.$$

We replace $\alpha$ by $\alpha(t)$ also in Eqs. (13–14).

In TOT, a tissue surrounding the wound is enclosed in a device with high oxygen pressure. The pressure increases the oxygen concentration to 5 times the normal concentration directly in the wound, independently of the ischemic condition; treatment is given daily for 1.5 hours [61]. We modify Eq. (9) as follows (Fig 5 and 6):

$$\frac{\partial W}{\partial t} - \nabla \cdot (vW) - \delta_W \nabla^2 W = \left(A_W + \lambda_{WV} \frac{V}{K_V + V}\right)(1 - \alpha) - d_W(F + M + E)W + \gamma_W h(t),$$ (20)

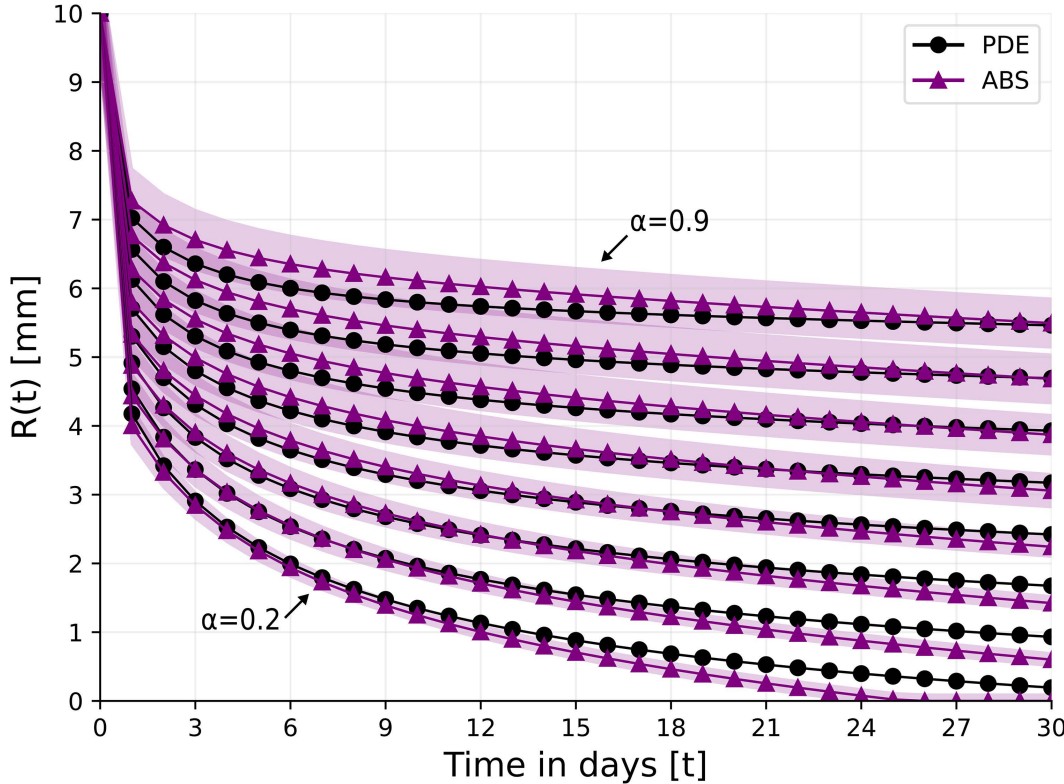

**Fig 5.** $\gamma_W^0$ = 3.

where $\gamma_W = 5A_W$ and

$$h(t) = \begin{cases} 1, & \text{if } 13 < t < 14.5 \text{ hours} \\ 0, & \text{elsewhere} \end{cases} \quad .$$

From Fig 2 we see that in the case of very mild ischemia where $\alpha$ = 0.1, wound closure is nearly complete by day 30. In order to simulate treatment with HBOT where $\alpha(t)$ is actually decreasing, we consider the cases where $\alpha \geq 0.2$.

Fig 7 shows the profile of $R(t)$ under HBOT treatment for two different treatments.

We see that under a small oxygen pressure of $\gamma_W^0$ = 1.5, wound closure is achieved for $\alpha$ = 0.2 after 28 days, but closure is not achieved in the ischemic case $\alpha$ = 0.3. On the other hand, with $\gamma_W^0$ = 3, wound closure for $\alpha$ = 0.2 is complete after 20 days, and in the case $\alpha$ = 0.3 it is nearly complete by day 30.

TCOT is a topical oxygen therapy given continuously 24 hours a day [61,62]. TCOT is used as adjunctive therapy in hard-to-heal wounds such as diabetic foot ulcer and pressure source ulcer; it provides a continuous supply of oxygen to promote healing. Here we shall consider the effect of TOT and TCOT on the closure of ischemic wounds. Figs 8 and 9 show the profiles of $R(t)$ under treatment with TOT and TCOT, respectively. With TOT treatment, wound closure in the case $\alpha$ = 0.2 is complete by day 21, but, for $\alpha$ = 0.3, it is only 95% complete by day 30. On the other hand, with TCOT wound closure is achieved (by day 25) for an ischemic level of $\alpha$ = 0.5.

When the ischemic level is high, namely, when $\alpha$ > 0.5, oxygen therapy cannot achieve wound closure in expected time. Non-healing wounds, such as highly ischemic wounds (e.g., with $\alpha$ > 0.5) are treated with debridement to remove

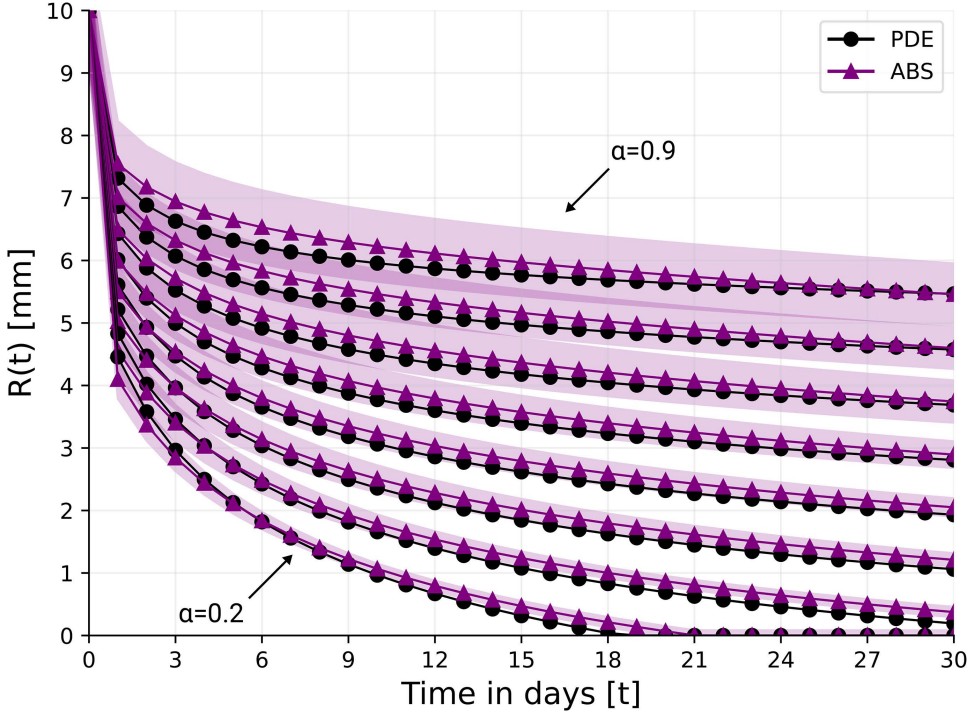

**Fig 6.** $\gamma_W^0 = 1.5$.

damaged tissue and prevent infection, and with compression therapy to help move blood around. In rare cases, non-healing wounds present a risk of life-threatening infection and may require amputation.

## 5. Discussion

### 5.1. Comparing PDE and ABS simulation results

In a PDE model of a biological process, the variables (species) are densities of cells, proteins, and other molecules at each point in space. In ABS model in 3D or (2D) space is covered with a uniform grid of size $\Delta$, cubes are of volume $\Delta^3$ (or $\Delta^2$), and each cube (or square) is occupied by at most one cell. In PDE models, the dynamics of the species is continuous in time, while in ABS, a set of rules is given, and simulations proceed in discrete time in a stochastic-probabilistic fashion. When the biological process takes place in a region whose boundary is unknown, the PDE system of equations must be complemented by a dynamic equation of the unknown boundary; this is not the case in the corresponding ABS model, where the boundary is automatically generated as cells proliferate and fill space.

Each of these two methods has its advantages and deficiencies. Hence it is interesting to compare their simulation results, particularly if we take the parameters associated with the rules of the ABS model from the parameters that appear to represent similar rules in the PDE model. This is what we did in the present paper, on ischemic wound closure. We found, quite surprisingly, that the boundaries of the open wound, $r = R(t)$, in the control case and under various oxygen treatments, as simulated by PDE and by ABS, are in very good agreement.

### 5.2. Minimal PDE model

In this paper, we developed a mathematical model to study the closure of ischemic wounds with or without oxygen therapy. Since there is always uncertainty in estimating the model parameters, one should aim for a model that has a

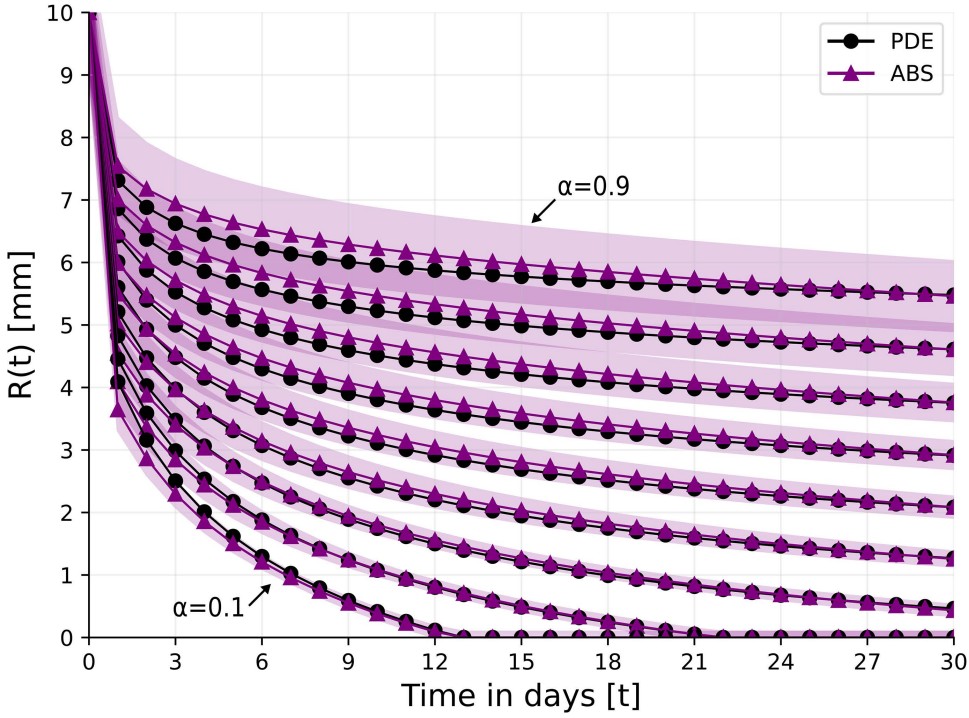

**Fig 7. Wound's radius over the course of 30 days for different levels of ischemia under two HBOT treatments.** For the ABS model, the results are shown as the mean ± standard deviation of n = 100 simulations.

"minimal" number of variables: the biological species should be those that are absolutely needed to simulate correctly the process of wound closure; species that are thought to affect wound closure rather marginally should be excluded. The decisions of what to include and what to exclude are a judgment call. In our model, we included fibroblasts, M2 macrophages, and keratinocytes, but not other epithelial cells, and not M1 cells. We also include VEGF, oxygen, and TGF-$\beta$ but explicitly PDGF. We also did not include TGF-$\beta$ and PDGF drugs since our focus was on ischemic wounds, and for the same reason, we did not include lipid molecules that play a role in chronic wounds and in age-associated wounds.

## 6. Conclusion

In this paper, we considered the healing of ischemic wounds, and focused on the proliferative phase, when the open wound is shrinking. We introduced a new PDE model of radially symmetric "flat" wound, which includes the primary role of keratinocyte cells, which make up to 90% of the cells of the epidermis. The radius $r = R(t)$ of the open wound at time $t$ is decreasing, and factors released from the area of the open wound ($A(t) = \pi R^2(t)$) increase the proliferation of fibroblasts and $M2$ macrophages.

We defined the level of ischemia by a parameter $\alpha$, that determines the flow rate of oxygen into the wound (Eq. (13)); $0 \leq \alpha \leq 1$, $\alpha = 0$ means no-ischemia while $\alpha = 1$ means total ischemia. In order to compute $R(t)$, we assumed, as in [1], that ECM is moving with velocity $v$ during the proliferation phase, and all species, including the wound boundary, are moving with the same velocity. Furthermore, we derived an equation for $v$, in terms of $\rho(t)$, by assuming that the tissue in $R(t) \leq r \leq R(0)$ has the viscous structure of a quasi-static upper convected Maywell fluid.

We also introduced another, very different, ABS model. In this model, the wound boundary $r = R(t)$ is generated automatically as cells proliferate in a stochastic-probabilistic manner. The rules of movement in ABS are entirely different from the rules in the PDE model, but we derived some of the parameters in the ABS model from appropriate parameters in the PDE model.

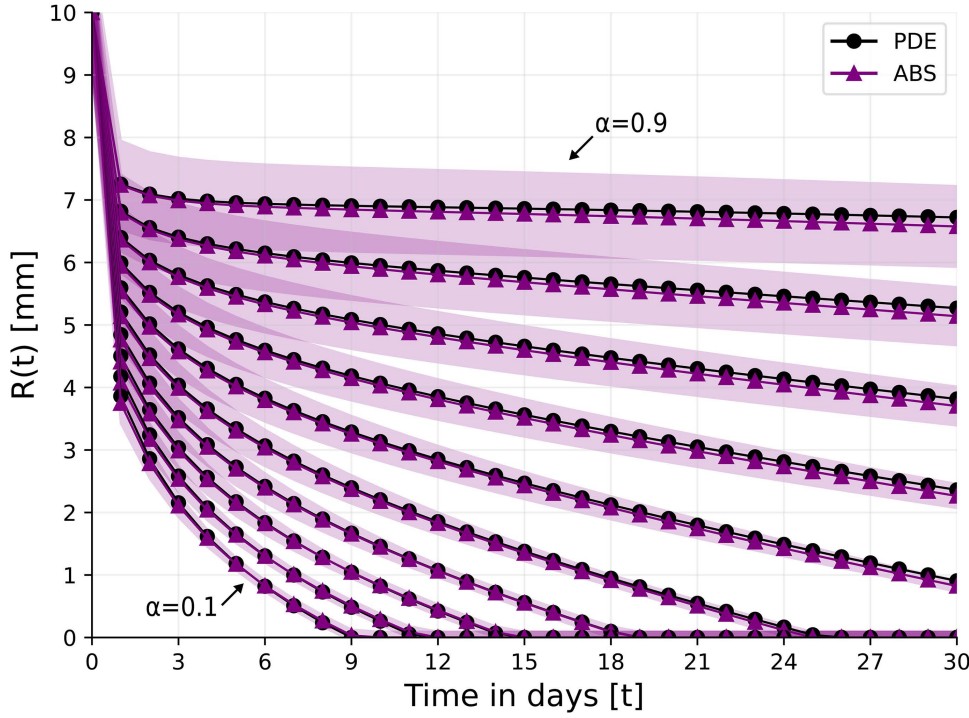

**Fig 8. Wound's radius over the course of 30 days for different levels of ischemia, $\alpha$ = 0.1, 0.2,…, 0.9 under the TOT treatment.** For the ABS model, the results are shown as the mean ± standard deviation of $n$ = 100 simulations.

We assumed that it takes at most 30 days to achieve closure of normal healthy wounds, and considered the question of in-time wound closure for ischemic wounds under various oxygen therapies. We obtained the following results:

(a) In the non-ischemic and ischemic cases, the model simulations are in good agreement with *in vivo* experiments made on domestic white pig [56]; the fitness measure is $R^2$=0.945 in the non-ischemic case and $R^2$=0.892 in the ischemic case (Fig 4).

(b) In all simulations of the model with $\alpha$ = 0, 0.1, 0.2,…, 0.9, the average measure of goodness of fit between the curves $R(t)$ in the PDE model and in the ABS model is $R^2$=0.913, which is surprisingly good; see also Discussion, sub-section 5.1.

(c) Treatment with HBOT can achieve wound closure in 30 days if $\alpha \leq 0.3$ (Fig 5), while treatment with TCOT can achieve closure if $\alpha \leq 0.5$ (Fig 7). If $\alpha > 0.5$, additional interventions will be needed.

The PDE model has the following limitations:

(1) The parameter $\alpha$ has not been mapped into a biologically measured value, such as oxygen pressure ($PO_2$) on the skin. Future in vivo experiments, such as [56], that also measure $PO_2$ in the wound environment could help provide a mapping for the model parameters $\alpha$ to $PO_2$.

(2) The average thickness of the epidermis is 0.1 cm, and the species (variables) in the PDE model are defined as densities in units of $g/cm^3$. But in the definition of the velocity $v$ (Eqs. (1–2)) and in all other model's equations we tacitly assumed, for simplicity, that the dynamics of wound closure does not depend on the thickness of the epidermis, thus treating the wound as a "flat" wound. The same simplification was introduced in the ABS model.

(3) As explained in Discussion sub-section 5.2, we developed, what we consider to be, a minimal model. This model still has many parameters, listed in Table 2. Some of the parameters are known from previous papers, while for a few

**Table 2. Summary of parameter estimations with their values.**

| Parameter | Description | Value | Reference |
|---|---|---|---|
| $E^0$ | Keratinocyte density in epidermis | $4.8 \cdot 10^{-2}$ $g/cm^3$ | estimated |
| $F^0$ | Fibroblast density in epidermis | $2.4 \cdot 10^{-3}$ $g/cm^3$ | estimated |
| $M^0$ | Macrophage density in epidermis | $2.4 \cdot 10^{-3}$ $g/cm^3$ | estimated |
| $V^0$ | VEGF density in skin | $1.5 \cdot 10^{-7}$ $g/cm^3$ | [38] |
| $T_\beta^0$ | $T_\beta$ density in skin | $3.5 \cdot 10^{-8}$ $g/cm^3$ | [39] |
| $W^0$ | Oxygen concentration in tissue | $4 \cdot 10^{-6}$ $g/cm^3$ | [40] |
| $\rho_0$ | ECM density in tissue | $0.06$ $g/cm^3$ | [41] |
| $\rho_m$ | Carrying capacity of $\rho$ | $0.066$ $g/cm^3$ | this work |
| $\rho_1$ | Threshold of internal pressure by $\rho$ | $0.012$ $g/cm^3$ | this work |
| $\beta$ | Growth parameter of internal pressure by $\rho$ | $2.72 \cdot 10^{-3}/d$ | this work |
| $d_F$ | Death rate of fibroblasts | $0.02/d$ | [43] |
| $d_M$ | Death rate of macrophages | $0.099/d$ | [44] |
| $d_E$ | Death rate of keratinocytes | $0.0577/d$ | [45] |
| $d_W$ | Consumption rate of $W$ by cells | $0.2/d$ | estimated |
| $d_V$ | Degradation rate of VEGF | $16.5/d$ | [46] |
| $d_{T_\beta}$ | Degradation rate of $T_\beta$ | $495/d$ | [47] |
| $d_\rho$ | Degradation rate of ECM | $0.37/d$ | [1] |
| $\delta_F, \delta_M, \delta_E$ | Diffusion coefficients of fibroblasts, macrophages, and keratinocytes | $8.64 \cdot 10^{-7}$ $cm^2/d$ | [48] |
| $\delta_G$ | Diffusion coefficient of VEGF | $8.64 \cdot 10^{-2}$ $cm^2/d$ | [49] |
| $\delta_W$ | Diffusion coefficient of oxygen | $2$ $cm^2/d$ | [50] |
| $\delta_{T_\beta}$ | Diffusion coefficient of $T_\beta$ | $7.1 \cdot 10^{-2}$ $cm^2/d$ | [51] |
| $K_V$ | Half-saturation of $G$ | $1.5 \cdot 10^{-7} g/cm^3$ | estimated |
| $K_{T_\beta}$ | Half-saturation of $T_\beta$ | $3.5 \cdot 10^{-8} g/cm^3$ | estimated |
| $F_0$ | Carrying capacity of $F$ | $4.8 \cdot 10^{-3} g/cm^3$ | estimated |
| $E_0$ | Carrying capacity of $E$ | $9.6 \cdot 10^{-2} g/cm^3$ | estimated |
| $A_F$ | Source of $F$ | $4.8 \cdot 10^{-5} g/cm^3$ | estimated |
| $A_M$ | Source of $M$ | $23.76 \cdot 10^{-2} g/cm^3$ | estimated |
| $\lambda_E$ | Growth rate of $E$ | $23.08 \cdot 10^{-2} g/cm^3$ | estimated |
| $\lambda_{VF}$ | Production of $V$ by $F$ | $1.031 \cdot 10^{-3}/d$ | estimated |
| $\lambda_{VM}$ | Production of $V$ by $M$ | $1.031 \cdot 10^{-3}/d$ | estimated |
| $\hat{d}_V$ | Loss of $V$ in the process of angiogenesis | $4.95 \cdot 10^{-6} g/cm^3 \cdot d$ | estimated |
| $A_W$ | Source of oxygen in tissue | $5.28 \cdot 10^{-2} g/cm^3 \cdot d$ | estimated |
| $\lambda_{WV}$ | Production of $W$ by VEGF | $11.616 \cdot 10^{-3} g/cm^3 \cdot d$ | estimated |
| $\lambda_{T_\beta F}$ | Production of $T_\beta$ by $F$ | $3.608/d$ | estimated |
| $\lambda_{T_\beta M}$ | Production of $T_\beta$ by $M$ | $3.608/d$ | estimated |
| $\lambda_{\rho T_\beta}$ | Enhanced production of $\rho$ by $T_\beta$ | $1$ | this work |
| $\lambda_\rho$ | Production of $\rho$ by $F$ | $1.356 \cdot 10^2/d$ | estimated |
| $\lambda_F$ | Production coefficient of $F$ associated with $A(t)$ | $8.154 \cdot 10^{-2}/cm^2 \cdot d$ | estimated |
| $\lambda_M$ | Production coefficient of $M$ associated with $A(t)$ | $7.14 \cdot 10^{-2}/cm^2 \cdot d$ | estimated |
| $\lambda_{MT_\beta}$ | $T_\beta$ enhanced production of $M$ by $T_\beta$ | $1$ | this work |

parameters there is no reference at all, and the chosen values are marked by "this work" in Table 2. The remaining parameters are derived from known experimental results either directly or under some "mild" assumption, as explained in Section 2.3, and they are marked by "estimated" in Table 2.

(4) When a wound occurs, it undergoes a process of stretching where its radius grows for several days, before it begins to decrease, as seen in [56] Fig 3. This process is not included in our model.

Mathematical models can be useful when they suggest new directions for research and experiments. When a mapping between the ischemic parameter $\alpha$ and $PO_2$ is developed as outlined in model's limitation (1), the PDE model could then be useful in suggesting personally optimal oxygen treatment for patients, based on their specific oxygen pressure.

The cells of the dermis include fibroblasts, macrophages, adipocytes, mast cells, Schwann cells, and stem cells [63], and, in deep wound healing, cells from both the epidermis and dermis start proliferating and migrating to the wound bed to close the wound [64]. In this paper, we consider the proliferation phase and wound closure of the epidermis. It would be interesting to extend the results of the paper to ischemic wounds deep into the dermis.

## Author contributions

**Conceptualization:** Avner Friedman.

**Data curation:** Avner Friedman.

**Formal analysis:** Teddy Lazebnik, Avner Friedman.

**Investigation:** Teddy Lazebnik, Avner Friedman.

**Methodology:** Teddy Lazebnik, Avner Friedman.

**Software:** Teddy Lazebnik.

**Validation:** Avner Friedman.

**Visualization:** Teddy Lazebnik.

**Writing – original draft:** Avner Friedman.

**Writing – review & editing:** Teddy Lazebnik, Avner Friedman.

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
