## [Decision Letter · Decision Letter 0]

16 Jul 2025

PONE-D-25-30699PDE and Agent Based Simulation Approaches to Ischemic Dermal Wound HealingPLOS ONE

Dear Dr. Lazebnik,

Thank you for submitting your manuscript to PLOS ONE. After careful consideration, we feel that it has merit but does not fully meet PLOS ONE’s publication criteria as it currently stands. Therefore, we invite you to submit a revised version of the manuscript that addresses the points raised during the review process.

If applicable, we recommend that you deposit your laboratory protocols in protocols.io to enhance the reproducibility of your results. Protocols.io assigns your protocol its own identifier (DOI) so that it can be cited independently in the future. For instructions see: https://journals.plos.org/plosone/s/submission-guidelines#loc-laboratory-protocols. Additionally, PLOS ONE offers an option for publishing peer-reviewed Lab Protocol articles, which describe protocols hosted on protocols.io. Read more information on sharing protocols at . Additionally, PLOS ONE offers an option for publishing peer-reviewed Lab Protocol articles, which describe protocols hosted on protocols.io. Read more information on sharing protocols at https://plos.org/protocols?utm_medium=editorial-email&utm_source=authorletters&utm_campaign=protocols..

We look forward to receiving your revised manuscript.

Kind regards,

Ahmed E. Abdel Moneim

Academic Editor

PLOS ONE

Journal Requirements:

https://journals.plos.org/plosone/s/file?id=ba62/PLOSOne_formatting_sample_title_authors_affiliations.pdf..

2. Please update your submission to use the PLOS LaTeX template. The template and more information on our requirements for LaTeX submissions can be found at http://journals.plos.org/plosone/s/latex..

3. Please note that PLOS One has specific guidelines on code sharing for submissions in which author-generated code underpins the findings in the manuscript. In these cases, we expect all author-generated code to be made available without restrictions upon publication of the work. Please review our guidelines at https://journals.plos.org/plosone/s/materials-and-software-sharing#loc-sharing-code and ensure that your code is shared in a way that follows best practice and facilitates reproducibility and reuse.

6. Please upload a copy of Figure 7, to which you refer in your text on page 8. If the figure is no longer to be included as part of the submission please remove all reference to it within the text.

7. Please include a copy of Table 3 which you refer to in your text on page 8.

Reviewers' comments:

Reviewer's Responses to Questions

**Comments to the Author**

1. Is the manuscript technically sound, and do the data support the conclusions?

Reviewer #1: Partly

Reviewer #2: No

Reviewer #3: Yes

Reviewer #4: Yes

2. Has the statistical analysis been performed appropriately and rigorously? 

Reviewer #1: Yes

Reviewer #2: N/A

Reviewer #3: Yes

Reviewer #4: N/A

3. Have the authors made all data underlying the findings in their manuscript fully available?

Reviewer #1: Yes

Reviewer #2: No

Reviewer #3: Yes

Reviewer #4: Yes

4. Is the manuscript presented in an intelligible fashion and written in standard English?

Reviewer #1: Yes

Reviewer #2: Yes

Reviewer #3: Yes

Reviewer #4: Yes

5. Review Comments to the Author

Reviewer #1: This study compares both partial differential equation modeling and agent-based modeling to analyze ischemic skin wound healing and further attempts to predict the therapeutic efficacy of oxygen therapy. I commend the authors for trying such a complex and clinically relevant issue using complementary computational methods. However, I believe the manuscript has some major issues that need to be addressed. While I am not a specialist in computational modeling, I offer the following comments from the perspective of biological interpretation, physiological plausibility, and clinical applicability.

1. Lack of experimental validation:

The simulation results regarding oxygen therapy are intriguing. However, to establish the credibility of the model, it is essential to first demonstrate that the model output under control (i.e., untreated) conditions is consistent with the natural course of wound healing observed in vivo. Without in vivo validation of the control scenario, conclusions regarding treatment efficacy remain speculative. Demonstrating that the model’s wound radius dynamics resemble experimental wound healing data should be the highest priority for ensuring the model's credibility.

2. Biological correlation of the parameter α:

The authors concluded that the efficiency of oxygen therapy is related to the parameter α. However, they did not provide a correspondence between α and physiological indices, such as TcPO₂. If α ≤ 0.5 serves as a critical threshold for predicting treatment effectiveness, the authors should provide at least an approximate mapping to biologically measurable values. If the simulation under control scenario is shown to be consistent with in vivo data, a practical conversion between the α range and clinical pathophysiology may become more feasible. Therefore, this issue may be partially addressed by resolving point 1.

3. Definition of keratinocyte density and criteria for wound closure:

In the models, wound closure appears to be assessed based on keratinocyte density, which is expressed in units of g/cm². This unit may not accurately reflect the biological process of epidermal layer formation or the functional closure of the wound. In particular, it is known that the epidermis immediately after re-epithelialization is thin, and thus the g/cm² value would remain lower than that of intact skin even after complete closure. However, if the simulated wound closure under control conditions matches in vivo healing trajectories, the practical validity of this unit definition and closure threshold can also be reinforced. Thus, this issue also relates back to the experimental validation discussed in point 1.

I have done my best to review the manuscript; however, as I am not a specialist in this specific field, I strongly recommend that an expert in mathematical or computational modeling also be consulted to provide a more rigorous evaluation.

Reviewer #2: 1.The research topic is quite meaningful, but from the reader's perspective, it is difficult to understand the author's intended meaning when reading the manuscript. It is also unclear how the mathematical models described by the author can guide clinical evaluation and wound management?

2.Question 1: Can so many and complex equations really evaluate the effectiveness of oxygen therapy in wound closure at different levels of ischemia? There are already many literature reports abroad that use transcutaneous oxygen pressure to measure the oxygen concentration in the surrounding tissues of wounds to evaluate the possibility of predicting wound healing. Wounds with an oxygen pressure level of 40mmHg or above indicate that wounds can heal, while wounds with an oxygen pressure level of 20-30mmHg can only partially heal and require hyperbaric oxygen therapy or topycal oxygen therapy. Wounds with an oxygen pressure level below 20mmHg need to rebuild blood supply and combine oxygen therapy for possible healing. This objective non-invasive measurement method can be repeatedly and dynamically measured without increasing the patient's pain, with accurate results, easy to operate and learn, and more in line with clinical needs. Why use such complex equations to evaluate?

3.Question 2: How to obtain and explain the result obtained by the author that "standard hyperbaric oxygen and local oxygen therapy can effectively achieve complete wound closure within the expected time when the ischemia level is not too high (i.e., α ≤ 0.5)"? How long does the expected time refer to? Are these results calculated through mathematical models? How to guide clinical wound assessment

4.Question 3: Some of the parameter estimates in Table 2 are based on references, while the majority are estimated. What is the basis for the estimated values? Readers also cannot understand.

5.Question 4: Wound healing is a very complex biological process, influenced by complex factors such as blood supply at the wound site, infection status, systemic nutrition, chronic disease comorbidities, psychological status, immune function, etc. Mathematical models, no matter how precise, cannot predict all unknown factors. So the possibility of evaluating and predicting wound healing must be comprehensively evaluated and analyzed by the patient's side in order to make a judgment.

6.Question 5: These two mathematical models have not been used for real wound assessment verification. Are the results of the author's predictions of wound healing true and reliable?

7. The discussion section is too simple, without discussing the scientific rationality and clinical practicality of the mathematical model, nor discussing the consistency between the obtained results and the clinical wound assessment results. What are the limitations?

Reviewer #3: the manuscript is well design and presented.

1) the Author has given all possible cells, tissue remodel proteins involved in the chronic wound healing model. why? did not include the lipid molecules. it is present 70% in cell membrane.

2) there is lot type error, sorry I lost tracking it. Please fix it.

3) Great mathematic chronic wound healing model. the Author should explain/showed experimental application.

Reviewer #4: Unfortunately, I am not qualified to review the details of mathematical modeling. Therefore, I focus more on the applicability of the results and the context and how it can advance the field of wound healing.

- How do the results correspond to in vivo observations? The authors should compare their results to the model wounds in rodent or porcine models of ischemic wounds and control nonischemic wounds.

- What are the applications of the model? What other asset do the models offer compared to in vivo ischemic models? I.e. what modalities or treatments can be tested? What would topical administration of PDGF or TGFb do with the wound

- The number of keratinocyte layers seems to be too high according to histological stainings. What was the source of this estimation?

- Also, keratinocytes migrate first as a single layer, then proliferate and differentiate. Basically part of the wound from the edges is covered with already differentiating epidermis, while the epithelial tongue comprises single or a few layers of cells.

- How does the number of cells change during the simulations? How does it correspond to in vivo situation?

- What are the limitations of this study?

- Which other parameters can be added to the model?

- How do the levels of growth factors change in your models? How does it relate to clinical wound healing? I did not find the concentration of VEGF in the wound in the referenced manuscript [37]. Was it derived?

- The manuscript [37] considers effects of VEGF besides angiogenesis, for example macrophages and keratinocyte migration. Was it somehow included in the models?

- Keratinocytes readily migrate in vitro in scratch assays. There are no other cells, or extrinsic factors beside those produced by keratinocytes or added to the medium (can be even without serum). Could the model capture this situation?

6. PLOS authors have the option to publish the peer review history of their article (what does this mean?). If published, this will include your full peer review and any attached files.). If published, this will include your full peer review and any attached files.

.

Reviewer #1: No

Reviewer #2: No

Reviewer #3: **Yes:** Dr. Bhagwat AlapureDr. Bhagwat Alapure

Reviewer #4: No

While revising your submission, please upload your figure files to the Preflight Analysis and Conversion Engine (PACE) digital diagnostic tool, https://pacev2.apexcovantage.com/. PACE helps ensure that figures meet PLOS requirements. To use PACE, you must first register as a user. Registration is free. Then, login and navigate to the UPLOAD tab, where you will find detailed instructions on how to use the tool. If you encounter any issues or have any questions when using PACE, please email PLOS at . PACE helps ensure that figures meet PLOS requirements. To use PACE, you must first register as a user. Registration is free. Then, login and navigate to the UPLOAD tab, where you will find detailed instructions on how to use the tool. If you encounter any issues or have any questions when using PACE, please email PLOS at figures@plos.org. Please note that Supporting Information files do not need this step.. Please note that Supporting Information files do not need this step.

---

## [Author Response · Author response to Decision Letter 1]

6 Sep 2025

Response to Reviewer #1

Review #2: “ This study compares both partial differential equation modeling and agent-based modeling to analyze ischemic skin wound healing and further attempts to predict the therapeutic efficacy of oxygen therapy. I commend the authors for trying such a complex and clinically relevant issue using complementary computational methods. However, I believe the manuscript has some major issues that need to be addressed. While I am not a specialist in computational modeling, I offer the following comments from the perspective of biological interpretation, physiological plausibility, and clinical applicability.“

We thank the reviewer for the thoughtful comments. We respond below to all his/her specific comments. We accordingly made many changes in the manuscript, which appear in RED color.

Comment #1: “The simulation results regarding oxygen therapy are intriguing. However, to establish the credibility of the model, it is essential to first demonstrate that the model output under control (i.e., untreated) conditions is consistent with the natural course of wound healing observed in vivo. Without in vivo validation of the control scenario, conclusions regarding treatment efficacy remain speculative. Demonstrating that the model’s wound radius dynamics resemble experimental wound healing data should be the highest priority for ensuring the model's credibility.“

We thank the reviewer for this comment. We added at the end of Section 4.1 a paragraph on the agreement of our simulations, under control, with in vivo experimental results on wound closure that appear in reference [55].a paragraph on the agreement of our simulations, under control, with in vivo experimental results on wound closure that appear in reference [55].

Comment #2: “The authors concluded that the efficiency of oxygen therapy is related to the parameter α. However, they did not provide a correspondence between α and physiological indices, such as TcPO₂. If α ≤ 0.5 serves as a critical threshold for predicting treatment effectiveness, the authors should provide at least an approximate mapping to biologically measurable values. If the simulation under control scenario is shown to be consistent with in vivo data, a practical conversion between the α range and clinical pathophysiology may become more feasible. Therefore, this issue may be partially addressed by resolving point 1.”

We thank the reviewer for this comment. We listed in the Conclusion section the lack of mapping between the parameter α and physiological index of PO2 as a model limitation (1); we also specifically noted in the results of the paper (item (c)) cases of effective treatment when α ≤ 0.3 and when α ≤ 0.5.

Comment #3: “In the models, wound closure appears to be assessed based on keratinocyte density, which is expressed in units of g/cm². This unit may not accurately reflect the biological process of epidermal layer formation or the functional closure of the wound. In particular, it is known that the epidermis immediately after re-epithelialization is thin, and thus the g/cm² value would remain lower than that of intact skin even after complete closure. However, if the simulated wound closure under control conditions matches in vivo healing trajectories, the practical validity of this unit definition and closure threshold can also be reinforced. Thus, this issue also relates back to the experimental validation discussed in point 1.”

We thank the reviewer for pointing the need to clarify the units. The units in the model are g/cm^3. In model limitation (2) of the Conclusion we explain that, for simplicity, we assumed that the dynamics of the wound closure does not depend on the thickness of the epidermis. So, mathematically, the equations take place in a plane, that is, the model is “flat”.

Response to Reviewer #2

“The research topic is quite meaningful, but from the reader's perspective, it is difficult to understand the author's intended meaning when reading the manuscript. It is also unclear how the mathematical models described by the author can guide clinical evaluation and wound management?”

We thank the reviewer for the thoughtful comments. We respond below to all his/her specific comments. We accordingly made many changes in the manuscript, which appear in RED color.

Comment #1: “Can so many and complex equations really evaluate the effectiveness of oxygen therapy in wound closure at different levels of ischemia? There are already many literature reports abroad that use transcutaneous oxygen pressure to measure the oxygen concentration in the surrounding tissues of wounds to evaluate the possibility of predicting wound healing. Wounds with an oxygen pressure level of 40mmHg or above indicate that wounds can heal, while wounds with an oxygen pressure level of 20-30mmHg can only partially heal and require hyperbaric oxygen therapy or topycal oxygen therapy. Wounds with an oxygen pressure level below 20mmHg need to rebuild blood supply and combine oxygen therapy for possible healing. This objective non-invasive measurement method can be repeatedly and dynamically measured without increasing the patient's pain, with accurate results, easy to operate and learn, and more in line with clinical needs. Why use such complex equations to evaluate?“

We thank the reviewer for this comment. In the Conclusion section, we listed, as the first model limitation, that we do not have a mapping of the ischemic parameter α to oxygen pressure level. Such a mapping could be developed when in vivo experiments as in [55] are done for ischemic wounds at different oxygen pressure. The model could then be useful in suggesting a personally optimal oxygen treatment for patients with any oxygen pressure. As for the “complex equations”, we explain in section 5.2 of the Discussion why we believe our model is actually “minimal”.of the Discussion why we believe our model is actually “minimal”.

Comment #2: “How to obtain and explain the result obtained by the author that "standard hyperbaric oxygen and local oxygen therapy can effectively achieve complete wound closure within the expected time when the ischemia level is not too high (i.e., α ≤ 0.5)"? How long does the expected time refer to? Are these results calculated through mathematical models? How to guide clinical wound assessment”

We were more precise now in the abstract to write: “not too high; i.e., α ≤ 0.5” in the abstract by the two lines marked in RED.

Comment #3: “Some of the parameter estimates in Table 2 are based on references, while the majority are estimated. What is the basis for the estimated values? Readers also cannot understand.”

We thank the reviewer for this comment. We address it in the Conclusion, by item (3) of model limitation.

Comment #4: “ Wound healing is a very complex biological process, influenced by complex factors such as blood supply at the wound site, infection status, systemic nutrition, chronic disease comorbidities, psychological status, immune function, etc. Mathematical models, no matter how precise, cannot predict all unknown factors. So the possibility of evaluating and predicting wound healing must be comprehensively evaluated and analyzed by the patient's side in order to make a judgment.”

We agree with the reviewer’s comment. We accordingly added a short paragraph in the Conclusion section, just after the list of model limitations.

Comment #5: “These two mathematical models have not been used for real wound assessment verification. Are the results of the author's predictions of wound healing true and reliable?”

We thank the reviewer for this comment. We now clarified in section 5.1 of the Discussion, that since both models, PDE and ABS, have their own advantages and deficiencies, it was interesting and worthwhile to compare if they yield the same results; our simulations show that they yield approximately the same results.of the Discussion, that since both models, PDE and ABS, have their own advantages and deficiencies, it was interesting and worthwhile to compare if they yield the same results; our simulations show that they yield approximately the same results.

Comment #6: “The discussion section is too simple, without discussing the scientific rationality and clinical practicality of the mathematical model, nor discussing the consistency between the obtained results and the clinical wound assessment results. What are the limitations?”

We thank the reviewer for this comment. We have written a new Discussion section and a mostly new Conclusion section that included the main results and limitations of the paper. The first limitation is that we do not have a mapping of the parameter α with the biologically measured oxygen pressure. Following the list of model’s limitation, we also added the following paragraph:

“Mathematical models can be useful when they suggest new directions for research and experiments. When a mapping between the ischemic parameter α and PO2 is developed as outlined in model's limitation (1), the PDE model could then be useful in suggesting personal optimally oxygen treatment for patients, based on their specific oxygen pressure.”

Response to Reviewer #3

We thank the reviewer for the thoughtful comments. We respond below to all his/her specific comments. We accordingly made many changes in the manuscript, which appear in RED color.

Comment #1: “the Author has given all possible cells, tissue remodel proteins involved in the chronic wound healing model. why? did not include the lipid molecules. it is present 70% in cell membrane.“

We thank the reviewer for this comment. In the Discussion, section 5.2 on minimal modeling, we explained that although lipid molecules play a role in chronic wounds and age-associated wounds, we did not include them in our study of ischemic wounds.on minimal modeling, we explained that although lipid molecules play a role in chronic wounds and age-associated wounds, we did not include them in our study of ischemic wounds.

Comment #2: “there is lot type error, sorry I lost tracking it. Please fix it.”

We corrected the type errors.

Comment #3: “Great mathematic chronic wound healing model. the Author should explain/showed experimental application.”

We thank the reviewer for this comment. In the new Conclusion section, we listed explicitly the results of the paper and model limitations. The first limitation is that we were unable to map the ischemic parameter α to a biological measure, like PO2. In the paragraph following the list of model limitations, we stated that as this mapping develops, the model could be useful in planning personally optimal oxygen treatment.

Response to Reviewer #4

“Unfortunately, I am not qualified to review the details of mathematical modeling. Therefore, I focus more on the applicability of the results and the context and how it can advance the field of wound healing.”

We thank the reviewer for the thoughtful comments. We respond below to all his/her specific comments. We accordingly made many changes in the manuscript, which appear in RED color.

Comment #1: “How do the results correspond to in vivo observations? The authors should compare their results to the model wounds in rodent or porcine models of ischemic wounds and control nonischemic wounds.“

We thank the reviewer for this comment. We added at the end of section 4.1 a paragraph on the agreement of our simulations in the control case with in vivo experimental results on wound closure, reported in reference [55].a paragraph on the agreement of our simulations in the control case with in vivo experimental results on wound closure, reported in reference [55].

Comment #2: “What are the applications of the model? What other asset do the models offer compared to in vivo ischemic models? I.e. what modalities or treatments can be tested? What would topical administration of PDGF or TGFb do with the wound”

Our response to these questions are as follows:

The application of the model can be summarized by the short paragraph in the Conclusion section that follows just after the list of the model limitations.

The PDE model is unique in including the primary role of keratinocyte cells in wound closure.

As explained in the Discussion, section 5.2, we developed a “minimal” model, and hence did not include PDGF and TGF-β therapies in this model of ischemic wounds; we included only oxygen therapies., we developed a “minimal” model, and hence did not include PDGF and TGF-β therapies in this model of ischemic wounds; we included only oxygen therapies.

Comment #3: “The number of keratinocyte layers seems to be too high according to histological stainings. What was the source of this estimation?”

The number of keratinocyte layers was based on references [34] and [7], as explained in the second paragraph of section 2.3.1..

Comment #4: “Also, keratinocytes migrate first as a single layer, then proliferate and differentiate. Basically part of the wound from the edges is covered with already differentiating epidermis, while the epithelial tongue comprises single or a few layers of cells.”

As explained in the Conclusion, model limitation (2), we assume, for simplicity, that the dynamics of the wound closure does not depend on the epidermis structure.

Comment #5: “How does the number of cells change during the simulations? How does it correspond to in vivo situation?”

The PDE equations are for density of cells, not for individual cells, they indicate how these densities move at each point in space and time; the model does not consider movement of the number of cells.

Comment #6: “What are the limitations of this study?”

In the Conclusion section, we listed the limitations (1)-(3) of the study.

Comment #7: “Which other parameters can be added to the model?”

As explained in the Discussion, section 5.2, we aimed to develop a “minimal” model. For this reason, we excluded additional species and parameters that, in our judgement, would not significantly affect the results of the study., we aimed to develop a “minimal” model. For this reason, we excluded additional species and parameters that, in our judgement, would not significantly affect the results of the study.

Comment #8: “How do the levels of growth factors change in your models? How does it relate to clinical wound healing? I did not find the concentration of VEGF in the wound in the referenced manuscript [37]. Was it derived?”

Since our focus was on wound closure, we presented the simulations of only the wound boundary R(t) and the keratinocyte average densities, and not the other species, such as the growth factors. The concentration of VEGF in the control case is given in Fig. 7 of reference [37].

Comment #9: “The manuscript [37] considers effects of VEGF besides angiogenesis, for example macrophages and keratinocyte migration. Was it somehow included in the models?”

Since we aimed to develop a “minimal” model, as explained in section 5.2 of the Discussion, we included only the primary role of angiogenesis of VEGF in ischemic wounds and excluded the other effects of VEGF.of the Discussion, we included only the primary role of angiogenesis of VEGF in ischemic wounds and excluded the other effects of VEGF.

Comment #10: “Keratinocytes readily migrate in vitro in scratch assays. There are no other cells, or extrinsic factors beside those produced by keratinocytes or added to the medium (can be even without serum). Could the model capture this situation?”

Since we aimed to develop a “minimal” model as explained in the Discussion, section 5.2, we did not consider this issue of keratinocyte migration. Our model cannot capture this situation., we did not consider this issue of keratinocyte migration. Our model cannot capture this situation.

---

## [Decision Letter · Decision Letter 1]

13 Oct 2025

PONE-D-25-30699R1PDE and Agent Based Simulation Approaches to Ischemic Dermal Wound HealingPLOS ONE

Dear Dr. Lazebnik,

Thank you for submitting your manuscript to PLOS ONE. After careful consideration, we feel that it has merit but does not fully meet PLOS ONE’s publication criteria as it currently stands. Therefore, we invite you to submit a revised version of the manuscript that addresses the points raised during the review process.

If applicable, we recommend that you deposit your laboratory protocols in protocols.io to enhance the reproducibility of your results. Protocols.io assigns your protocol its own identifier (DOI) so that it can be cited independently in the future. For instructions see: https://journals.plos.org/plosone/s/submission-guidelines#loc-laboratory-protocols. Additionally, PLOS ONE offers an option for publishing peer-reviewed Lab Protocol articles, which describe protocols hosted on protocols.io. Read more information on sharing protocols at . Additionally, PLOS ONE offers an option for publishing peer-reviewed Lab Protocol articles, which describe protocols hosted on protocols.io. Read more information on sharing protocols at https://plos.org/protocols?utm_medium=editorial-email&utm_source=authorletters&utm_campaign=protocols..

We look forward to receiving your revised manuscript.

Kind regards,

Ahmed E. Abdel Moneim

Academic Editor

PLOS ONE

Journal Requirements:

Reviewers' comments:

Reviewer's Responses to Questions

**Comments to the Author**

1. If the authors have adequately addressed your comments raised in a previous round of review and you feel that this manuscript is now acceptable for publication, you may indicate that here to bypass the “Comments to the Author” section, enter your conflict of interest statement in the “Confidential to Editor” section, and submit your "Accept" recommendation.

Reviewer #1: (No Response)

Reviewer #2: All comments have been addressed

Reviewer #4: (No Response)

2. Is the manuscript technically sound, and do the data support the conclusions?

Reviewer #1: Partly

Reviewer #2: Yes

Reviewer #4: Partly

3. Has the statistical analysis been performed appropriately and rigorously? 

Reviewer #1: Yes

Reviewer #2: (No Response)

Reviewer #4: N/A

4. Have the authors made all data underlying the findings in their manuscript fully available?

Reviewer #1: Yes

Reviewer #2: Yes

Reviewer #4: Yes

5. Is the manuscript presented in an intelligible fashion and written in standard English?

Reviewer #1: Yes

Reviewer #2: Yes

Reviewer #4: Yes

6. Review Comments to the Author

Reviewer #1: While the authors have attempted to address my previous concerns, the core issues regarding validation against experimental data remain unresolved. The revisions do not provide the level of rigor necessary for publication. Please find my detailed comments below.

Major Concerns

#1. The comparison with in vivo data (Ref. [55]) is superficial. It is limited to a qualitative visual inspection of figures, without any quantitative validation. No comparison method is described in the Methods, and the Results lack the rigor required for scientific validation. Moreover, the assignment of ischemic level α = 0.5 to represent the ischemic model in Ref. [55] is arbitrary. In addition, the re-epithelialization data are inconsistent: in Ref. [55], non-ischemic wounds show ~60% closure at Day 7, whereas the model predicts only ~20% (α = 0). Therefore, the model and the experimental data are not consistent.The authors emphasize that the two mathematical models (PDE and ABS) are in close agreement with each other. However, what matters is agreement with experimental reality, not merely internal consistency between two simplified models. Agreement between models that both deviate from in vivo data does not strengthen credibility.

#2. The limitation regarding the lack of mapping between the ischemic parameter α and physiological indices remains unresolved. Ref. [55] reports SPP values for each wound model, but since the present study’s predictions are already inconsistent with the experimental wound closure data (see comment#1), no meaningful mapping between α and clinical indices can be established.

The origin of this discrepancy with in vivo results is not entirely clear. It may stem from parameter inaccuracies, or from oversimplified modeling assumptions—for example, the flat “2D epidermis” assumption (regarding the comment #3). Such simplifications may critically distort wound healing dynamics. While further refinement of the model may eventually address these issues, in its current form the model does not achieve biological plausibility.

Because the manuscript’s central claim of experimental consistency is unsupported, and the models fail to reproduce established in vivo wound healing data, the study does not provide reliable or clinically translatable predictions. I therefore recommend rejection of this version.

Reviewer #2: Oxygen therapy for ischemic wounds has been under clinical research, but there are many doubts about how to evaluate its effectiveness.Sincerely appreciate the authors for making substantial revisions based on the reviewers' comments, clarifying many uncertainties, and significantly enhancing the discussion section with additional and improved content.

Reviewer #4: Thank you for addressing the comments. The manuscript is much clearer and more applicable. To further support your conclusions, please discuss HBOT vs TCOT results. Is the difference in efficacy of the two methods comparable to those of in vivo?

Does your model generate some hypotheses that can be tested in vivo?

7. PLOS authors have the option to publish the peer review history of their article (what does this mean?). If published, this will include your full peer review and any attached files.). If published, this will include your full peer review and any attached files.

.

Reviewer #1: No

Reviewer #2: No

Reviewer #4: No

While revising your submission, please upload your figure files to the Preflight Analysis and Conversion Engine (PACE) digital diagnostic tool, https://pacev2.apexcovantage.com/. PACE helps ensure that figures meet PLOS requirements. To use PACE, you must first register as a user. Registration is free. Then, login and navigate to the UPLOAD tab, where you will find detailed instructions on how to use the tool. If you encounter any issues or have any questions when using PACE, please email PLOS at . PACE helps ensure that figures meet PLOS requirements. To use PACE, you must first register as a user. Registration is free. Then, login and navigate to the UPLOAD tab, where you will find detailed instructions on how to use the tool. If you encounter any issues or have any questions when using PACE, please email PLOS at figures@plos.org. Please note that Supporting Information files do not need this step.. Please note that Supporting Information files do not need this step.

---

## [Author Response · Author response to Decision Letter 2]

3 Nov 2025

Dear editors and reviewers,

We are pleased and privileged to resubmit our manuscript “PDE and Agent Based Simulation Approaches to Ischemic Dermal Wound Healing" for your reconsideration to be published in Ploe One.

We would like to thank the reviewers for their kind words and detailed review. We believe that after addressing the comments, the paper is much clearer and better structured for potential readers. In particular, we paid special attention to showing the model capture in vivo data.

To make the review process as smooth as possible, we highlight all the additions and edits in the manuscript in red and blue fonts. In addition, a formal cover letter detailing our responses to the reviewers’ comments point-by-point is attached below.

Kind regards,

The authors

Reviewer #1:

Comment: “The comparison with in vivo data (Ref. [55]) is superficial. It is limited to a qualitative visual inspection of figures, without any quantitative validation. No comparison method is described in the Methods, and the Results lack the rigor required for scientific validation. Moreover, the assignment of ischemic level α = 0.5 to represent the ischemic model in Ref. [55] is arbitrary. In addition, the re-epithelialization data are inconsistent: in Ref. [55], non-ischemic wounds show ~60% closure at Day 7, whereas the model predicts only ~20% (α = 0). Therefore, the model and the experimental data are not consistent.The authors emphasize that the two mathematical models (PDE and ABS) are in close agreement with each other. However, what matters is agreement with experimental reality, not merely internal consistency between two simplified models. Agreement between models that both deviate from in vivo data does not strengthen credibility. The limitation regarding the lack of mapping between the ischemic parameter α and physiological indices remains unresolved. Ref. [55] reports SPP values for each wound model, but since the present study’s predictions are already inconsistent with the experimental wound closure data (see comment#1), no meaningful mapping between α and clinical indices can be established. The origin of this discrepancy with in vivo results is not entirely clear. It may stem from parameter inaccuracies, or from oversimplified modeling assumptions—for example, the flat “2D epidermis” assumption (regarding the comment #3). Such simplifications may critically distort wound healing dynamics. While further refinement of the model may eventually address these issues, in its current form the model does not achieve biological plausibility. Because the manuscript’s central claim of experimental consistency is unsupported, and the models fail to reproduce established in vivo wound healing data, the study does not provide reliable or clinically translatable predictions. I therefore recommend rejection of this version.”

Response: Thank the reviewer for these comments, and we accordingly made changes in the manuscript in RED.

The reviewer is correct in, and we thank him for, criticising our sloppy assertion about the agreement between our model’s simulations and [55]. We have now performed correctly the simulation of wound closure, noting that wound closure in [55] Fig. 3 is given not by R(t) but by the percentage of R(t)/R(0) and that R(0) = 0.2 in [55]) (not R(0) = 1cm in our model). The simulation results now fit well with [55] Fig. 3 as seen in the new Fig. 4B.

Note that we started these simulations at day 3 since our model does not include the process of initial wound stretching, which takes place initially for several days (as noted in item (4) of the model limitation, page 20).

With respect to epithelialization in [55] Fig. 4 for days 2 and 7, we note that the results in [55] were derived by mason tricome stain over the entire wound tissue, while our results on keratincyte density (in Fig. 3) are taken in the region where R(t) < r < R(0). Furthermore, and more importantly, the initial stretching of the wound (which is not included in our model) significantly affects epithelialization. We therefore do not expect keratinocyte density in our model (with radius of percentage of R(t)/R(0), and with R(0) = 0.2cm) to fit the epithelialization in [55] Fig. 4; hence, this is not included in the present version.

Reviewer #2:

Comment: “Oxygen therapy for ischemic wounds has been under clinical research, but there are many doubts about how to evaluate its effectiveness.Sincerely appreciate the authors for making substantial revisions based on the reviewers' comments, clarifying many uncertainties, and significantly enhancing the discussion section with additional and improved content.”

Response: Thank you for the positive comment.

Reviewer #4:

Comment: “To further support your conclusions, please discuss HBOT vs TCOT results. Is the difference in efficacy of the two methods comparable to those of in vivo? Does your model generate some hypotheses that can be tested in vivo?”

Response: We added a new paragraph (in Blue) on page 16. TCOT is not used in wounds that are just ischemic. But we think it is interesting to see what benefit it could potentially have compared to TOT.

---

## [Decision Letter · Decision Letter 2]

26 Nov 2025

PONE-D-25-30699R2PDE and Agent Based Simulation Approaches to Ischemic Dermal Wound ClosurePLOS ONE

Dear Dr. Lazebnik,

Thank you for submitting your manuscript to PLOS ONE. After careful consideration, we feel that it has merit but does not fully meet PLOS ONE’s publication criteria as it currently stands. Therefore, we invite you to submit a revised version of the manuscript that addresses the points raised during the review process.

If applicable, we recommend that you deposit your laboratory protocols in protocols.io to enhance the reproducibility of your results. Protocols.io assigns your protocol its own identifier (DOI) so that it can be cited independently in the future. For instructions see: https://journals.plos.org/plosone/s/submission-guidelines#loc-laboratory-protocols. Additionally, PLOS ONE offers an option for publishing peer-reviewed Lab Protocol articles, which describe protocols hosted on protocols.io. Read more information on sharing protocols at . Additionally, PLOS ONE offers an option for publishing peer-reviewed Lab Protocol articles, which describe protocols hosted on protocols.io. Read more information on sharing protocols at https://plos.org/protocols?utm_medium=editorial-email&utm_source=authorletters&utm_campaign=protocols..

We look forward to receiving your revised manuscript.

Kind regards,

Ahmed E. Abdel Moneim

Academic Editor

PLOS ONE

Journal Requirements:

Reviewers' comments:

Reviewer's Responses to Questions

**Comments to the Author**

1. If the authors have adequately addressed your comments raised in a previous round of review and you feel that this manuscript is now acceptable for publication, you may indicate that here to bypass the “Comments to the Author” section, enter your conflict of interest statement in the “Confidential to Editor” section, and submit your "Accept" recommendation.

Reviewer #1: (No Response)

Reviewer #4: All comments have been addressed

2. Is the manuscript technically sound, and do the data support the conclusions?

Reviewer #1: Partly

Reviewer #4: Yes

3. Has the statistical analysis been performed appropriately and rigorously? 

Reviewer #1: Yes

Reviewer #4: N/A

4. Have the authors made all data underlying the findings in their manuscript fully available?

Reviewer #1: Yes

Reviewer #4: Yes

5. Is the manuscript presented in an intelligible fashion and written in standard English?

Reviewer #1: Yes

Reviewer #4: Yes

6. Review Comments to the Author

Reviewer #1: The comparison between the present model and the in vivo data reported in Ref. 55 is now more convincing than in the previous version. The agreement demonstrated in Fig. 4 appears reasonable and provides a certain level of support for the model.

However, the methodology used for this comparison is not described in the Methods section. Instead, several methodological details appear only within the Results section, where Methods and Results are intermixed.

Once the comparison methodology is appropriately moved and clarified, I believe the manuscript will meet the standard for publication.

Reviewer #4: (No Response)

7. PLOS authors have the option to publish the peer review history of their article (what does this mean?). If published, this will include your full peer review and any attached files.). If published, this will include your full peer review and any attached files.

.

Reviewer #1: No

Reviewer #4: No

---

## [Author Response · Author response to Decision Letter 3]

28 Nov 2025

Attached a rebuttal letter to the submission

---

## [Decision Letter · Decision Letter 3]

23 Dec 2025

PDE and Agent Based Simulation Approaches to Ischemic Dermal Wound Closure

PONE-D-25-30699R3

Dear Dr. Lazebnik,

We’re pleased to inform you that your manuscript has been judged scientifically suitable for publication and will be formally accepted for publication once it meets all outstanding technical requirements.

An invoice will be generated when your article is formally accepted. Please note, if your institution has a publishing partnership with PLOS and your article meets the relevant criteria, all or part of your publication costs will be covered. Please make sure your user information is up-to-date by logging into Editorial Manager at Editorial Manager® and clicking the ‘Update My Information' link at the top of the page. For questions related to billing, please contact  and clicking the ‘Update My Information' link at the top of the page. For questions related to billing, please contact billing support..

Kind regards,

Ahmed E. Abdel Moneim

Academic Editor

PLOS One

Additional Editor Comments (optional):

Reviewers' comments:

Reviewer's Responses to Questions

**Comments to the Author**

1. If the authors have adequately addressed your comments raised in a previous round of review and you feel that this manuscript is now acceptable for publication, you may indicate that here to bypass the “Comments to the Author” section, enter your conflict of interest statement in the “Confidential to Editor” section, and submit your "Accept" recommendation.

Reviewer #1: All comments have been addressed

2. Is the manuscript technically sound, and do the data support the conclusions?

Reviewer #1: Yes

3. Has the statistical analysis been performed appropriately and rigorously? 

Reviewer #1: Yes

4. Have the authors made all data underlying the findings in their manuscript fully available?

Reviewer #1: Yes

5. Is the manuscript presented in an intelligible fashion and written in standard English?

Reviewer #1: Yes

6. Review Comments to the Author

Reviewer #1: (No Response)

7. PLOS authors have the option to publish the peer review history of their article (what does this mean?). If published, this will include your full peer review and any attached files.). If published, this will include your full peer review and any attached files.

.

Reviewer #1: No
